# What Matters in Deep Learning for Time Series Forecasting?

## Abstract

Deep learning models have grown increasingly popular in time series applications. However, the large quantity of newly proposed architectures, together with often contradictory empirical results, makes it difficult to assess which components contribute significantly to final performance. We aim to make sense of the current design space of deep learning architectures for time series forecasting by discussing the design dimensions and trade-offs that can explain, often unexpected, observed results. We discuss the necessity of grounding model design on principles for forecasting groups of time series and how such principles can be applied to current models. In particular, we assess how concepts such as locality and globality apply to recent forecasting architectures. We show that accounting for these aspects can be more relevant for achieving accurate results than adopting specific sequence modeling layers and that simple, well-designed forecasting architectures can often match the state of the art. We discuss how overlooked implementation details in existing architectures (1) fundamentally change the class of the resulting forecasting method and (2) drastically affect the observed empirical results. Our results call for rethinking current faulty benchmarking practices and for the need to focus on the foundational aspects of the forecasting problem when designing neural network architectures. As a step in this direction, we also propose an *auxiliary forecasting model card*, i.e., a template with a set of fields to characterize existing and new forecasting architectures based on key design choices.

## 1 Introduction

Novel sequence modeling architectures are consistently improving the state-of-the-art in many applications (Gu et al., 2022; Gu and Dao, 2024; Beck et al., 2024), such as text and natural language processing. However, results in time series forecasting offer a much more uncertain way ahead, with recent work questioning the effectiveness of modern deep learning approaches (Toner and Darlow, 2024; Zeng et al., 2023; Tan et al., 2024). The result is that current research is seemingly stuck in a loop of positive results being quickly dismissed by new evidence that questions our understanding of the components that contribute to obtaining accurate forecasts (Shao et al., 2024). Recent works propose several architectures, e.g., based on attention (Zhou et al., 2021; Wu et al., 2021; Nie et al., 2023; Liu et al., 2023a; Zhang and Yan, 2023; Liu et al., 2022a), and assess their performance against state-of-the-art methods on common benchmarks. Most of these architectures are obtained by stacking and combining different components and operators and involve many–often hidden–implementation choices (e.g., parameter sharing and local parameters). However, the impact of such design choices on the resulting model and its performance is often overlooked. As an example, in recent works, a collection of synchronous time series is often considered as a single multivariate signal. This approach can lead to misconceptions and results that are difficult to interpret. Starting from this consideration, recent architectures stemming from Nie et al. (2023) rely on what has been called *channel-independence*, i.e., on processing each channel of a time series independently from others while sharing the same parameters. This approach has–somewhat surprisingly–led to superior results when compared to standard multivariate models. However, when these channels correspond to different univariate and homogeneous (related) time series (as is often the case in commonly used benchmarks), "channel-independence" corresponds to adopting the framework of global models, which is well understood in time series analysis (Benidis et al., 2022; Salinas et al., 2020; Januschowski et al., 2020; Montero-Manso and Hyndman, 2021). While this

might sound simply an issue of naming conventions, it can provide clear explanations for observed results (Montero-Manso and Hyndman, 2021) and unlock new designs (Wang et al., 2019; Smyl, 2020). For example, there is a large body of literature on methods that account for dependencies among synchronous time series while keeping (part of) the model global (Cini et al., 2023; Sen et al., 2019). As an additional example, other works have recently observed that applying attention among channels can improve performance (Liu et al., 2023a; Zhang and Yan, 2023). However, if we consider the different channels as a collection of correlated time series, similar attention operators have been routinely used in spatiotemporal forecasting models (Ma et al., 2019; Grigsby et al., 2021; Marisca et al., 2022; Liu et al., 2023b). Besides missed opportunities and insights, these aspects can also harm the effectiveness of our benchmarking practices. Indeed, overlooked design choices can, as we will show, lead to empirical results that are difficult to interpret and that might mislead the designer. Moreover, as we will discuss throughout the paper, those mentioned are only a selection of the issues that contribute to the current situation.

In this paper, we scan the design space of modern deep learning architectures for time series forecasting and assess the impact that associated design choices have on current benchmarking practices. In particular, we aim at understanding the state of the field by relying on well-understood principles for forecasting groups of time series. In doing so, we examine recent architectures and empirical results, highlighting the impact of overlooked aspects that are often considered as mere implementation details. To frame the discussion, we structure our analysis by considering four main dimensions: **D1.**) **model configuration**, i.e., selecting the model family (e.g., local, global, or hybrid); **D2.**) **preprocessing and exogenous variables**, i.e., selecting exogenous variables and setting up preprocessing and postprocessing operations; **D3.**) **temporal processing**, i.e., accounting for temporal (i.e., intra-series) dependencies. **D4.**) **spatial processing**, i.e., accounting for spatial (i.e., inter-series) dependencies. While some of these dimensions are not always orthogonal (e.g., space and time contributions can be processed in an integrated way), we believe that analyzing how these different aspects concur to characterize a model family is the key to understanding recent results. We argue that, to assess meaningful improvements to the state of the art, any comparison must ensure that design choices in any of these dimensions do not interfere with the evaluation of the proposed component. In this context, our contributions are as follows.

- We analyze the current state of deep learning for time series forecasting by relying on principles to forecast groups of time series to make sense of often contradictory empirical results.

- We empirically assess the impact of overlooked design choices and implementation details in existing state-of-the-art architectures, and show that they explain a significant portion of the observed performance improvements.

- We show that a streamlined architecture built on well-understood design principles can match the performance of the state-of-the-art.

- To move forward, we introduce an *auxiliary forecasting model card* template–complementary to existing generic model cards (Mitchell et al., 2019)–that can be used to characterize existing and new forecasting architectures.

The current trends in the field have led to focusing on finding a one-size-fits-all architecture with state-of-the-art performance in benchmarks. This prompted the adoption of increasingly more complex architectures that combine many poorly understood components. By showing the limitations of common benchmarking practices, our work is aimed at stimulating discussion on our current approach to conducting machine learning for time series forecasting. We believe that this discussion is an important step for the maturity of the field and to ensure future progress.

## 2 RELATED WORK AND CONTEXT

The history of neural networks in forecasting applications is long, and has often been characterized by skepticism (Zhang et al., 1998). However, the forecasting community is reaching consensus on the effectiveness of deep learning methods in settings where a single neural network can be trained on (large) collections of related time series (Hewamalage et al., 2021; Benidis et al., 2022). Models based on this approach have been called *global* in contrast with *local* models, which are instead trained separately on each time series (Montero-Manso and Hyndman, 2021; Januschowski et al., 2020; Benidis et al., 2022). Global models and hybrid global-local variants thereof have won

forecasting competitions (Smyl, 2020) and been adopted by the industry (Salinas et al., 2020; Kunz et al., 2023). With the increase in popularity of new sequence modeling architectures (Vaswani et al., 2017; Gu et al., 2022; Gu and Dao, 2024; Orvieto et al., 2023), the machine learning community has started investigating how to adapt such architectures to the forecasting problem. In particular, the Informer (Zhou et al., 2021) architecture is among the first of a line of works aiming at tailoring Transformers (Vaswani et al., 2017) to long-range time series forecasting. Together with the architecture, Zhou et al. (2021) also introduced a popular benchmark where collections of time series are considered as a single multivariate sequence. Several subsequent works follow, then, the same approach (Wu et al., 2021; Liu et al., 2022a; Wu et al., 2023; Liu et al., 2022b; Zhou et al., 2022). Zeng et al. (2023) and Toner and Darlow (2024) show that most of these architectures can be outperformed in such benchmarks by simple linear models. Nie et al. (2023), then, showed that–in the same settings–superior results could be achieved by processing each channel independently with shared parameters. For many of these benchmarks, this essentially corresponds to the global approach; indeed, a large part of the associated datasets consists of collections of related time series, even though, as already mentioned, they have often been seen as a single multivariate sequence. Follow-up works (Liu et al., 2023a; Zhang and Yan, 2023) then reintroduced components to model dependencies across multiple time series while keeping the core of the model global. Conflating the problem of forecasting any group of time series into forecasting a single multivariate sequence, as we will see, can be problematic and lead to unclear designs (Sec. 4.1). Moreover, current popular architectures stack several components and rely on many hidden implementation choices, which make a direct comparison of introduced sequence modeling operators challenging.

The need to clarify inconsistencies in benchmarking practices has pushed researchers to focus on developing new benchmarks and evaluation pipelines for forecasting (Shao et al., 2024; Wang et al., 2024a; Qiu et al., 2024). Conversely, in our work, we aim to assess whether the performance gains of the recently proposed method come from the use of specific operators or lie in other implementation details. Said differently, we aim to assess whether current benchmarking practices are focusing on *what really matters* in deep learning for time series forecasting. Similar analysis has been done in other subfields of machine learning, such as reinforcement learning (Raichuk et al., 2021), and in the context of graph neural networks (GNNs) (Errica et al., 2020; Dwivedi et al., 2023).

## 3 PRELIMINARIES

### 3.1 PROBLEM SETTING

We consider a collection of $N$ time series $\mathcal{D} = \{\boldsymbol{x}_{0:L_1}^1, \ldots, \boldsymbol{x}_{0:L_N}^N\}$, where $\boldsymbol{x}_{0:L_i}^i \in \mathbb{R}^{L_i \times d_x}$ denotes the sequence of $L_i$ $d_x$-dimensional observations associated with the $i$-th time series. When present, exogenous variables are denoted as $\boldsymbol{u}_{0:L_i}^i \in \mathbb{R}^{L_i \times d_u}$. A binary mask, $\boldsymbol{m}_{0:L_i}^i \in \{0,1\}^{L_i \times d_x}$, may be introduced to model missing or invalid observations. Time series in the set can come from different domains and be generated by different stochastic processes. In such a setting, the mask $\boldsymbol{m}_{0:L_i}^i$ can be used to account for heterogeneous time series by modeling missing channels. If time series are synchronous, we use capital letters to denote values across the collection, e.g., $\boldsymbol{X}_t \in \mathbb{R}^{N \times d_x}$ refers to the stacked observations at time step $t$. Time series in the collection might be *correlated* (in a broad sense), i.e., uncertainty on future values of each time series might be reduced by taking into account observations from other time series.

**Forecasting groups of time series** We consider the problem of multi-step ahead time series forecasting, i.e., the problem of predicting the next $H \geq 1$ observations $\boldsymbol{x}_{t:t+H}^i$ for the i-th time series, given a window $W \geq 1$ of past observations $\boldsymbol{x}_{t-W:t}^i$ from the same time series. As the stochastic process generating data $p^i$ is unknown, the objective is to approximate it with a model $p_{\boldsymbol{\theta}}$ with learnable parameters $\boldsymbol{\theta}$ such that

$$p_{\boldsymbol{\theta}}(\boldsymbol{x}_{t:t+H}^i \mid \boldsymbol{x}_{t-W:t}^i, \boldsymbol{u}_{t-W:t}^i, \boldsymbol{u}_{t:t+H}^i) \approx p^i(\boldsymbol{x}_{t:t+H}^i \mid \boldsymbol{x}_{<t}^i, \boldsymbol{u}_{<t}^i, \boldsymbol{u}_{t:t+H}^i) \quad \forall i = 1, \ldots, N \quad (1)$$

where $\boldsymbol{x}_{<t}^i$ denotes all past observations of the $i$-th series preceding timestep $t$. We focus on the problem of obtaining *point forecasts* $\widehat{\boldsymbol{x}}_{t:t+H}^i$ of, e.g., the expected value such as $\widehat{\boldsymbol{x}}_{t:t+H}^i \approx \mathbb{E}_p\left[\boldsymbol{x}_{t:t+H}^i\right]$ by using a parametric model $\mathcal{F}(\,\cdot\,; \boldsymbol{\theta})$. Predictions are obtained by fitting parameters $\boldsymbol{\theta}$ of the chosen model family. As we will discuss in Sec. 4.1, we say that a model is *global* if its parameters are shared across all the time series. In such a case, the model is trained on the entire set of

time series. Conversely, a model is *local* if its parameters are specific to a single time series. If relying on local models, forecasting a collection of time series results in fitting a separate model for each sequence in the set. Choosing between a local and global approach (or a hybrid thereof) depends on the task at hand, data availability, and model complexity. As mentioned in Sec. 1, due to advantages in sample efficiency, global models are a particularly appealing choice when relying on deep learning architectures (Hewamalage et al., 2021; Benidis et al., 2022). Additionally, global models can be employed inductively, e.g., in a cold start scenario, whereas local models are transductive. We will expand this discussion in Sec. 4.1.

## 3.2 BASELINES

Through the paper, we assess the impact of different design choices with respect to a set of recent state-of-the-art architectures for long-range time series forecasting, which we compare against simpler, streamlined baselines (see Sec. 3.2). We consider representative models that have shaped the development of recent time series forecasting methods and that demonstrate competitive performance on benchmarks. We include: 1. **PatchTST** (Nie et al., 2023), the widely used architecture that–as already mentioned–introduced "channel independence" and patching-based Transformer layers. In particular, PatchTST relies on splitting the input into fixed-size patches before applying attention; 2. **DLinear** (Zeng et al., 2023), which combines a linear model with a time series decomposition step; 3. **TimeMixer** (Wang et al., 2024b), which is an multilayer perceptron (MLP)-based architecture processing the input at different resolutions; 4. **Linear**, a linear autoregressive model trained with $L2$ regularization and ordinary least squares (OLS), following Toner and Darlow (2024), implemented in both its global and local variants. We also consider models that incorporate spatial processing: 5. **iTransformer** (Liu et al., 2023a), which processes the temporal dynamics with a feedforward layer and then uses standard attention among channels; 6. **ModernTCN** (Donghao and Xue, 2024) which employs convolutional layers for spatio-temporal representation; 7. **Crossformer** (Zhang and Yan, 2023), which uses patching and spatiotemporal attention operators to model dependencies among different channels of the input time series. To ensure a fair comparison, we evaluate all the models in the same benchmarking setup, under unified settings, and with access to the same exogenous variables. We rely on the available open-source implementations of each approach and adapt them to our evaluation procedure and standardized inputs. The code for all the experiments, based on the Torch Spatiotemporal library (Cini and Marisca, 2022), will be open-sourced upon publication. For a more detailed description of each baseline, we refer the reader to App. A.

**Reference architectures**  In our experiments, we compare state-of-the-art models against a reference streamlined architecture specifically designed to assess the impact of different design choices w.r.t. the target design dimensions. Note that the purpose here is not to propose a new architecture to challenge the state of the art. Conversely, reference architectures provide baselines, introduced to facilitate a fair and consistent comparison and to gauge the impact of different design choices more directly. For the temporal module, we consider several alternatives: a MLP with residual connections, a temporal convolutional network (TCN) with causal dilated filters (Bai et al., 2018), a gated recurrent neural network (RNN) (Chung et al., 2014), a stack of Transform layers (Vaswani et al., 2017), and pyramidal attention operators akin to the Pyraformer architecture (Liu et al., 2022a). In the tables, we denote these reference models as **MLP**, **TCN**, **RNN**, **Transf.**, **Pyraf.**, respectively. For the TCN, RNN and attention-based models, we use a $1$-$D$ convolutional layer with a large stride as an additional preprocessing step to implement an operator akin to patching (Nie et al., 2023) and facilitate the processing at the subsequent layers. For the spatial module, we use a simple spatial attention layer (denoted as **sp. attn.**). For additional implementation details, please refer to App. B.

## 3.3 BENCHMARKS

As a benchmark, we use four real-world datasets from different domains that are widely used in the context of long-range time series forecasting (Wang et al., 2024b; Zhang and Yan, 2023; Liu et al., 2023a; Zeng et al., 2023; Nie et al., 2023). In particular: **Electricity** collects hourly electricity usage for 321 customers (Wu et al., 2021); **Weather** includes 21 meteorological variables collected every 10 minutes from Germany (Wu et al., 2021); **Traffic** contains hourly road occupancy data collected from various 862 sensors on San Francisco's highways (Wu et al., 2021); **Solar** contains 10-minute records of solar power generation from 137 photovoltaic plants (Lai et al., 2018). We use a $70\%/10\%/20\%$

split for the training, validation, and testing, following previous works (Wang et al., 2024b). Metrics are reported on scaled data for consistency with published benchmarks. All the numerical results are averaged over three independent runs. We use a window size of 96 for all experiments, while in Tab. 3 we use a longer window size of 336 (except for Solar). For further details, refer to App. C.

# 4 WHAT MATTERS IN DEEP LEARNING FOR TIME SERIES FORECASTING?

In this section, we go through four key design dimensions that characterize forecasting architectures and have a significant impact on overall performance. For each design dimension, we assess how different choices have contributed to making the current progress in the field uncertain, leading to often unexpected empirical results.

**D1. Model configuration** This refers to the type of forecasting model being employed. We distinguish between local models (each trained on individual time series), global models (trained on multiple series jointly with the same shared weights), and hybrid approaches that combine elements of both paradigms.

**D2. Preprocessing and exogenous variables** This dimension refers to the transformations applied to the data either before or after being used as input to a predictor, and to the exogenous variables used as additional inputs to the forecasting architecture.

**D3. Temporal processing** Temporal processing refers to the operators used to model temporal dependencies within an architecture.

**D4. Spatial processing** This dimension involves mechanisms used to model inter-series dependencies when multiple time series are available as inputs.

We do not aim to provide an exhaustive discussion of each dimension, but instead focus on how they have been addressed in recent research and highlight their impact on performance.

## 4.1 DESIGN DIMENSION 1: MODEL CONFIGURATION

As previously discussed, the model configuration–global, local, or hybrid –is a fundamental aspect in model design, since it radically changes the type of model being used. Yet, this aspect is often left unspecified or dealt with as an implementation detail. However, choosing between a local, global, or hybrid approach has several implications that should be properly discussed (Montero-Manso and Hyndman, 2021; Januschowski et al., 2020; Salinas et al., 2020). For instance, it is often common to model any collection of synchronous time series as a single highly-dimensional multivariate time series and hence consider models such as

$$\widehat{\boldsymbol{X}}_{t:t+H} = \mathcal{F}\left(\boldsymbol{X}_{t-W:t}, \ldots; \boldsymbol{\theta}\right). \tag{2}$$

However, this approach can scale poorly with the input's dimensionality. Indeed, in practice, several recent works (e.g., Nie et al. (2023); Liu et al. (2023a)) have observed that processing each channel independently with the same parameters empirically yields better performance. As already mentioned, this is equivalent to adopting the well-known global approach, i.e., to processing related time series as

$$\widehat{\boldsymbol{x}}^i_{t:t+H} = \mathcal{F}\left(\boldsymbol{x}^i_{t-W:t}, \ldots; \boldsymbol{\theta}\right) \quad \forall\, i = 1, \ldots, N. \tag{3}$$

Moreover–although not always explicitly stated–several architectures, e.g., (Wang et al., 2024b; Zhang and Yan, 2023; Donghao and Xue, 2024), adopt the approach in Eq. 3, but introduce some time series specific parameters $\boldsymbol{\phi}^i$, resulting in models

$$\widehat{\boldsymbol{x}}^i_{t:t+H} = \mathcal{F}\left(\boldsymbol{x}^i_{t-W:t}, \ldots; \boldsymbol{\theta}, \boldsymbol{\phi}^i\right) \quad \forall\, i = 1, \ldots, N. \tag{4}$$

which effectively consist of hybrid global-local models (Smyl, 2020; Cini et al., 2023; Benidis et al., 2022). The design choices that, in practice, lead to models as in Eq. 4 are often dealt with as implementation details. For instance, Wang et al. (2024b) uses learnable local parameters in the normalization module; Salinas et al. (2020)–while relying on an otherwise global model–uses a different one-hot-encoding vector associated with each processed time series, effectively introducing a vector of learnable parameters specific to that input sequence. Finally, an opposite trend seen in other approaches–often relying on simple (linear) models (Zeng et al., 2023)–design models in Eq. 2 by using different parameters for each time series

$$\widehat{\boldsymbol{x}}^i_{t:t+H} = \mathcal{F}\left(\boldsymbol{x}^i_{t-W:t}, \ldots; \boldsymbol{\theta}^i\right) \quad \forall\, i = 1, \ldots, N, \tag{5}$$

hence yielding to *local models*.

Clearly, models in Eq. 2–5 correspond to fundamentally different approaches that can result in markedly different performance. Failing to recognize the impact of the associated design choices can be problematic for several reasons. First, the use of shared versus local parameters may have very different effects depending on whether the time series are homogeneous (e.g., data coming from identical sensors at different locations) or heterogeneous (e.g., measurements of different physical quantities). Moreover, when dealing with multiple multivariate time series, a multivariate global model is often more appropriate than a univariate one that processes channels independently. Second, as we will see, comparing the results of models belonging to different typologies without stating it explicitly can make it difficult to interpret performance differences. In Tab. 1, we assess the performance–in terms of mean squared error (MSE)–of different architectures from the literature on a set of benchmarks (see Sec. 3.3). As in all our experiments, we focus on the task of long-range time series forecasting, with a prediction horizon of 96 time steps. We evaluate changes in performance for the reference Transformer and for two architectures that incorporate local parameters by removing these components, and, conversely, for the iTransformer by adding them.

Table 1: Comparison (MSE) of models with local embeddings. Best average results are in **bold**.

| D | Model | Hybrid | Global |
|---|---|---|---|
| Electr. | Transf. | $\mathbf{0.136}_{\pm.000}$ | $0.151_{\pm.000}$ |
| | Crossformer | $\mathbf{0.141}_{\pm.001}$ | $0.146_{\pm.003}$ |
| | TimeMixer | $\mathbf{0.151}_{\pm.000}$ | $0.180_{\pm.001}$ |
| | iTransformer | $\mathbf{0.139}_{\pm.000}$ | $0.154_{\pm.000}$ |
| Weather | Transf. | $\mathbf{0.153}_{\pm.001}$ | $0.177_{\pm.002}$ |
| | Crossformer | $\mathbf{0.154}_{\pm.003}$ | $0.164_{\pm.003}$ |
| | TimeMixer | $\mathbf{0.164}_{\pm.002}$ | $0.178_{\pm.001}$ |
| | iTransformer | $\mathbf{0.154}_{\pm.000}$ | $0.170_{\pm.001}$ |
| Traffic | Transf. | $0.417_{\pm.009}$ | $\mathbf{0.392}_{\pm.000}$ |
| | Crossformer | $0.540_{\pm.014}$ | $\mathbf{0.512}_{\pm.007}$ |
| | TimeMixer | $0.464_{\pm.001}$ | $\mathbf{0.463}_{\pm.001}$ |
| | iTransformer | $0.435_{\pm.002}$ | $\mathbf{0.409}_{\pm.000}$ |
| Solar | Transf. | $\mathbf{0.196}_{\pm.000}$ | $0.205_{\pm.001}$ |
| | Crossformer | $0.177_{\pm.008}$ | $\mathbf{0.166}_{\pm.005}$ |
| | TimeMixer | $\mathbf{0.366}_{\pm.017}$ | $0.367_{\pm.017}$ |
| | iTransformer | $\mathbf{0.189}_{\pm.001}$ | $0.197_{\pm.002}$ |

As one would expect, using local parameters drastically changes the observed results. Mixing results from the two columns without accounting for the impact of these design choices would clearly lead to misleading conclusions. Therefore, when aiming to identify the most effective sequence modeling operators, experiments should be designed to factor out the impact of model configuration.

## 4.2 DESIGN DIMENSION 2: PREPROCESSING AND EXOGENOUS VARIABLES

Exogenous variables and preprocessing (e.g., scaling, detrending, and methods accounting for seasonality) are ingredients that can have a significant impact on final performance. In this section, we discuss how exogenous variables and preprocessing methods have been considered and included inconsistently across popular baselines and benchmarks. Similar to model configuration, these benchmarking practices further prevent a clear understanding of the reasons behind the observed performances. This issue is compounded by the growing trend of comparing newly proposed architectures directly against published results of existing methods, without reproducing those results in this exercise. It follows that differences in preprocessing routines become increasingly difficult to isolate and account for. To investigate the extent of this problem in recent benchmarks, we focus specifically on the use of exogenous variables. For instance, PatchTST, DLinear, and Crossformer do not use covariates by default, while iTransformer does. We then evaluate the impact of adding the same covariates (calendar features, in this case) to some of these baselines and report the outcome of this experiment in Tab. 2. Results show the impact of including covariates on the performance of models such as DLinear, PatchTST and Crossformer, which do exclude them in their original implementations, as well as on iTransformer and the reference Transformer, which include them. Their effect is more pronounced on some

Table 2: Comparison (MSE) of models with and without covariates. Best average results are in **bold**

| D | Model | w/ exog. | w/out exog. |
|---|---|---|---|
| Electr. | Transf. | $\mathbf{0.136}_{\pm.000}$ | $0.155_{\pm.001}$ |
| | PatchTST | $\mathbf{0.128}_{\pm.000}$ | $0.134_{\pm.000}$ |
| | Crossformer | $\mathbf{0.139}_{\pm.002}$ | $0.141_{\pm.001}$ |
| | iTransformer | $\mathbf{0.154}_{\pm.000}$ | $0.167_{\pm.000}$ |
| | DLinear | $\mathbf{0.193}_{\pm.000}$ | $0.195_{\pm.000}$ |
| Weather | Transf. | $\mathbf{0.153}_{\pm.001}$ | $0.161_{\pm.000}$ |
| | PatchTST | $\mathbf{0.174}_{\pm.000}$ | $0.180_{\pm.002}$ |
| | Crossformer | $0.154_{\pm.003}$ | $0.154_{\pm.003}$ |
| | iTransformer | $\mathbf{0.170}_{\pm.001}$ | $0.176_{\pm.000}$ |
| | DLinear | $0.199_{\pm.005}$ | $\mathbf{0.196}_{\pm.001}$ |
| Traffic | Transf. | $\mathbf{0.417}_{\pm.009}$ | $0.479_{\pm.006}$ |
| | PatchTST | $\mathbf{0.355}_{\pm.000}$ | $0.383_{\pm.001}$ |
| | Crossformer | $0.548_{\pm.024}$ | $\mathbf{0.540}_{\pm.014}$ |
| | iTransformer | $\mathbf{0.409}_{\pm.000}$ | $0.444_{\pm.001}$ |
| | DLinear | $\mathbf{0.609}_{\pm.000}$ | $0.648_{\pm.000}$ |
| Solar | Transf. | $\mathbf{0.196}_{\pm.000}$ | $0.206_{\pm.003}$ |
| | PatchTST | $\mathbf{0.196}_{\pm.001}$ | $0.225_{\pm.003}$ |
| | Crossformer | $\mathbf{0.176}_{\pm.006}$ | $0.177_{\pm.008}$ |
| | iTransformer | $\mathbf{0.197}_{\pm.002}$ | $0.221_{\pm.003}$ |
| | DLinear | $\mathbf{0.246}_{\pm.001}$ | $0.285_{\pm.001}$ |

Table 3: Forecasting results (MSE and MAE) for a horizon of 96 steps for models *not including* spatial processing. Best average results are in **bold**, second best are underlined.

| MODEL | Electricity | | Weather | | Traffic | | Solar | |
|---|---|---|---|---|---|---|---|---|
| | MSE | MAE | MSE | MAE | MSE | MAE | MSE | MAE |
| Linear Global | 0.140 | 0.237 | 0.174 | 0.234 | 0.410 | 0.282 | 0.222 | 0.291 |
| Linear Local | 0.134 | 0.230 | **0.144** | 0.209 | 0.426 | 0.298 | 0.223 | 0.295 |
| MLP | 0.129$_{\pm.000}$ | 0.225$_{\pm.000}$ | 0.148$_{\pm.001}$ | 0.198$_{\pm.000}$ | 0.376$_{\pm.000}$ | 0.253$_{\pm.001}$ | 0.194$_{\pm.003}$ | 0.239$_{\pm.002}$ |
| RNN | 0.147$_{\pm.001}$ | 0.247$_{\pm.001}$ | 0.149$_{\pm.001}$ | 0.203$_{\pm.001}$ | 0.390$_{\pm.007}$ | 0.275$_{\pm.002}$ | 0.200$_{\pm.003}$ | 0.246$_{\pm.004}$ |
| TCN | 0.130$_{\pm.000}$ | 0.224$_{\pm.000}$ | 0.148$_{\pm.000}$ | 0.200$_{\pm.001}$ | 0.364$_{\pm.003}$ | 0.253$_{\pm.002}$ | 0.193$_{\pm.004}$ | 0.243$_{\pm.005}$ |
| Transf. | 0.129$_{\pm.001}$ | 0.222$_{\pm.001}$ | 0.149$_{\pm.001}$ | 0.203$_{\pm.002}$ | 0.362$_{\pm.003}$ | 0.249$_{\pm.002}$ | 0.203$_{\pm.006}$ | 0.245$_{\pm.002}$ |
| Pyraf. | 0.129$_{\pm.001}$ | 0.224$_{\pm.001}$ | 0.148$_{\pm.001}$ | 0.199$_{\pm.001}$ | 0.365$_{\pm.002}$ | 0.251$_{\pm.003}$ | **0.189**$_{\pm.003}$ | **0.236**$_{\pm.004}$ |
| TimeMixer | 0.129$_{\pm.001}$ | 0.224$_{\pm.000}$ | 0.147$_{\pm.001}$ | 0.197$_{\pm.000}$ | 0.373$_{\pm.002}$ | 0.271$_{\pm.003}$ | 0.199$_{\pm.001}$ | 0.245$_{\pm.000}$ |
| PatchTST | **0.125**$_{\pm.000}$ | **0.218**$_{\pm.000}$ | 0.148$_{\pm.001}$ | **0.195**$_{\pm.001}$ | **0.345**$_{\pm.000}$ | **0.234**$_{\pm.000}$ | 0.197$_{\pm.001}$ | 0.244$_{\pm.004}$ |
| DLinear | 0.140$_{\pm.000}$ | 0.237$_{\pm.000}$ | 0.173$_{\pm.000}$ | 0.232$_{\pm.001}$ | 0.407$_{\pm.000}$ | 0.283$_{\pm.000}$ | 0.246$_{\pm.001}$ | 0.331$_{\pm.000}$ |

benchmarks while less evident in others. These results pinpoint another source of uncertainty in interpreting recent benchmarking results; preprocessing steps should be standardized across baselines to ensure that all models have access to the same inputs.

### 4.3 DESIGN DIMENSION 3: TEMPORAL PROCESSING

In this section, we assess whether streamlined models, properly configured as hybrid global-local models with exogenous inputs and local embeddings, can achieve performance comparable to that of recent state-of-the-art models. This design dimension, concerning sequence modeling operators, has been the main focus of recent research. However, this line of work has produced contrasting results, leading to considerable confusion about which components effectively contribute to performance (Toner and Darlow, 2024; Zeng et al., 2023; Tan et al., 2024). We focus on methods that only process inputs along the temporal dimension, while approaches that include components accounting for spatial dependencies are discussed in Sec. 4.4. For this analysis, we use the different reference architectures introduced in Sec. 3.2, and compare them against three popular and well-established baselines, namely DLinear, PatchTST, and TimeMixer, by using standardized inputs (including covariates), and hyperparameter tuning. We remark that the goal is not to determine which architecture performs best, but rather to assess the extent to which different temporal processing components influence the observed results. As shown in Tab. 3, no single model consistently outperforms the others. Moreover, reference architectures that rely on standard and simple operators obtain competitive performance against the state of the art across all the considered scenarios. These results challenge the effectiveness of current benchmarking practices in identifying the components responsible for performance improvements and in measuring the contribution brought by the different sequence modeling operators. Additionally, results show that, in many scenarios, choosing a specific sequence modeling operator is not the critical design choice. Analogous observations are confirmed in Sec. 4.4.

Note that in all experiments, we process data from the Weather dataset as if it were a collection of univariate time series, to show the effect of handling it as is commonly done in the literature. Interestingly, one of the best-performing models on Weather is the local OLS linear model. This is not too surprising, since Weather is actually a multivariate time series with heterogeneous channels, and among the models in Tab. 3, that linear model is the only one that explicitly models each time series as heterogeneous. Although results do not provide a clear ordering of methods, we did observe that patching, which, as mentioned earlier, corresponds to applying a nonlinear convolutional filter over time, works well across both reference architectures and state-of-the-art baselines, providing a good approach for enabling the processing of long input windows. Hierarchical attention-based approaches (such as Pyraformer) also showed to be a viable option. Finally, we report in Fig. 1 an assessment of the computational scalability of the different architectures, in terms of time needed to process each batch and GPU memory usage. The computational cost is visualized in relation to the forecasting accuracy. Results show that MLP, TCN, and PatchTST achieve a good trade-off between MSE performance, GPU memory usage, and training time. We encourage conducting analysis like this to gain insight into the most suitable models for given benchmarks.

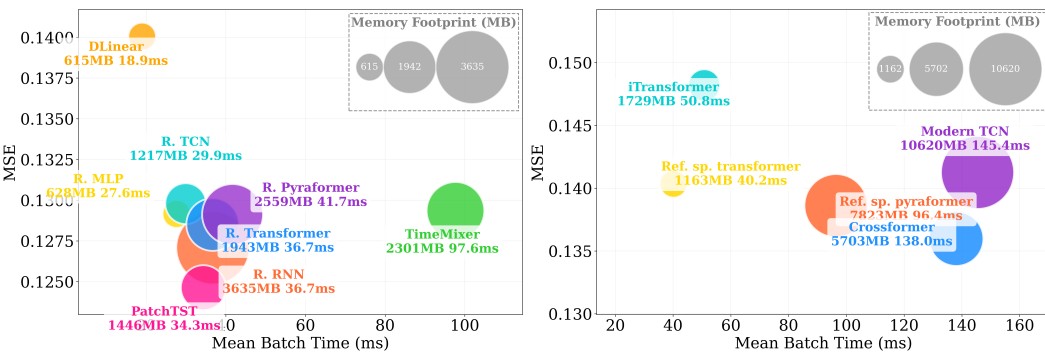

(a) Models *not including* spatial processing.

(b) Models *including* spatial processing.

Figure 1: MSE versus mean batch time during training on the Electricity dataset. Circle size indicates memory consumption.

Table 4: Forecasting results (MSE) for a horizon of 96 steps for models *including* spatial processing. Best average results are in **bold**, second best are underlined.

(a) Comparison of reference architectures with spatial attention against state-of-the-art baselines.

| Model | Electricity | Weather | Traffic | Solar |
|---|---|---|---|---|
| MLP + sp. attn. | $0.140_{\pm.001}$ | $0.157_{\pm.000}$ | $0.435_{\pm.006}$ | $0.201_{\pm.009}$ |
| Pyraf. + sp. attn. | $0.139_{\pm.001}$ | $0.157_{\pm.002}$ | $\mathbf{0.389_{\pm.002}}$ | $\underline{0.188_{\pm.002}}$ |
| iTransformer | $0.148_{\pm.000}$ | $0.171_{\pm.001}$ | $\underline{0.393_{\pm.001}}$ | $0.208_{\pm.003}$ |
| Crosformer | $\mathbf{0.136_{\pm.000}}$ | $\mathbf{0.152_{\pm.003}}$ | $0.527_{\pm.002}$ | $\mathbf{0.184_{\pm.008}}$ |
| ModernTCN | $0.141_{\pm.000}$ | $\underline{0.154_{\pm.001}}$ | $0.445_{\pm.001}$ | $0.190_{\pm.001}$ |

(b) **Ablation:** iTransformer with or without spatial attention.

| | iTransformer | |
|---|---|---|
| Dataset | Spatial att. | Feedforward |
| Electricity | $\mathbf{0.148_{\pm.000}}$ | $0.149_{\pm.001}$ |
| Weather | $\mathbf{0.171_{\pm.001}}$ | $\mathbf{0.171_{\pm.000}}$ |
| Traffic | $0.393_{\pm.001}$ | $\mathbf{0.390_{\pm.001}}$ |
| Solar | $0.208_{\pm.003}$ | $\mathbf{0.194_{\pm.001}}$ |

## 4.4 DESIGN DIMENSION 4: SPATIAL PROCESSING

We call *spatial* the dimension that spans multiple time series, which may correspond to different spatial locations when considering physical sensors. This section complements the discussion started in Sec. 4.3 by considering models that account for inter-series dependencies by relying on different operators. In particular, we compare the reference architectures, where dependencies are modeled with a standard spatial Transformer, against three state-of-the-art baselines: iTransformer, Crossformer, and ModernTCN. For the reference architectures, we use an MLP or pyramidal attention for temporal processing. Tab. 4a reports the results of the comparison where we reduced the length of the input window associated with each time series to keep computational costs manageable. Analogously to Sec. 4.3, simulations show that the simple, streamlined architectures perform comparably to the state of the art, highlighting once again the limitations of current benchmarking practices. The results in Tab. 3 and Tab. 4a, combined with the observation that spatial dependencies might provide limited benefits in long-range forecasting, have led us to doubt the effectiveness of spatial attention operators in this context. For this reason, we design an additional architecture by removing the spatial attention layer, replaced by a simple MLP, in iTransformer, effectively removing all the components modeling spatial dependencies in the architecture. Results in Tab. 4b indeed show that in this context, entirely removing spatial attention led to better or similar performance in all the considered datasets. These observations further highlight the need for more thorough assessments of how each component contributes to the observed results. Finally, Fig. 1 reports performance in relation to computational cost, which in this case is particularly critical as processing data along the spatial dimension can have a significant impact on computational scalability.

## 5 DISCUSSION

The results in Sec. 4.1–4.4 question whether we have been successful in measuring the advances of deep learning architectures for time series forecasting and shed light on several faults in current benchmarking practices. We showed that overlooked design choices can have a significant impact and that

simple, well-designed architectures can match the state of the art on standard benchmarks (see Tab. 3 and 4a). Our analysis calls for reaching a better understanding of the architecture's design space, showing how misconceptions in model specification can trigger misleading conclusions (as shown, e.g., in Tab. 1). The additional ablation studies (e.g., in Tab. 2 and 4b) corroborate these findings that are further substantiated by plenty of additional empirical results in the appendix of the paper. However, our objective is not to be dismissive of the progress of the field–which is tangible in many applications–but rather to ensure that we can move forward by focusing on answering foundational questions and fostering awareness of the existing flaws in our practices. Revisiting the benchmarking pipeline will be a crucial step in this direction. For instance, introducing benchmarks specifically designed to isolate distinct dimensions, possibly by relying on synthetic datasets, can help measure the isolated effects of different design choices. Moreover, model cards (Mitchell et al., 2019) can be an effective tool, providing a simple and practical way to summarize a model's main characteristics and to facilitate model comparison. Below, we propose a set of *forecasting model cards* that can be used in conjunction with existing model cards to capture relevant aspects of the design dimensions discussed in the paper, and we provide an example of its use in App. D. We believe that by recalibrating our evaluation tools on reliably measuring actual progress, the shift toward answering more foundational methodological questions would happen as a natural consequence.

---

### FORECASTING MODEL CARD

**Model setting**

- Size of the input window
- Whether the model is transductive or inductive, and can be used in a cold start scenario
- How to mask missing observations and/or if imputation is needed

**D1. Model configuration**

- Whether the model is global, local, or hybrid
- *If the model is hybrid*, which parameters are shared across the time series and which are not

**D2. Preprocessing and exogenous variables**

- The type of scaling or other transformation applied at training and inference time
- Temporal covariates, lagged variables, or other types of exogenous variables are employed

**D3. Temporal processing**

- Modules and operators used to encode observations along the temporal axis
- Time and space complexity w.r.t. the length of the time series being processed

**D4. Spatial processing**
*If spatial dependencies are accounted for:*

- Modules used to model spatial dynamics and whether a graph structure is employed
- Time and space complexity w.r.t. the number of the time series being processed

---

## 6 CONCLUSION

We investigated the effectiveness of current benchmarks in measuring progress in the field and the impact of design choices on forecasting performance. We showed, by analyzing points of failure in our evaluation procedures, that our current practices might produce misleading results. With this paper, we pinpointed several of the sources of this uncertainty and aimed to foster a discussion to ensure that the field can move forward and address its current limitations. Indeed, our analysis also shows that appropriate design choices do have an impact on performance and can explain seemingly contradictory empirical results. We believe that the results and the analysis presented in this paper are an important step toward focusing on *what matters* in deep learning for time series forecasting.

**Limitations** In this work, we focused on a specific set of dimensions in the design space of forecasting models that have had a strong impact on the benchmarking results of recent studies.

Clearly, this analysis can be extended to other aspects of the design space. For example, the discussion can be expanded to include probabilistic forecasting and the choice of metrics to quantify forecasting accuracy. Moreover, it could cover additional aspects included in the forecasting model card template, such as missing values, graph structures, and transductive versus inductive settings. Additionally, future work could explore similar issues in short-term forecasting benchmarks.

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

# APPENDIX

## A   BASELINES

Table 5: Description for the baseline models

| Model | Model configuration | Temporal processing | Spatial processing |
| --- | --- | --- | --- |
| Dlinear | Global | Linear layers | Not modeled |
| PatchTST | Global | Temporal convolution followed by temporal attention over the patches | Not modeled |
| TimeMixer | Hybrid | Feedforward networks applied to the trend and seasonal components, downsampled at different scales | Not modeled |
| Crossformer | Hybrid | Temporal convolution followed by attention applied over the patches, with a hierarchical structure constructed with linear layers | Spatial attention applied among patches of different time series |
| iTransformer | Global | Feedforward layers | Spatial attention applied among different time series |
| ModernTCN | Hybrid | Depth-wise convolutions | Convolution applied across time series |
| Linear global/local | Global/local | Linear autoregression | Not modeled |

Below, we provide a brief description of each baseline as employed in our experiments on the considered benchmarks. Furthermore, we summarize them in Tab. 5 using three fields corresponding

to the design dimensions introduced in Sec. 4, excluding the *preprocessing and exogenous variables* dimension, since the considerable differences among the methods make it less informative.

- Dlinear (Zeng et al., 2023) decomposes the input into seasonal and trend components using a moving average and processes them with linear layers. The hyperparameters determine its local-global nature. In the table, we report it as global because, in our experiments, it was used in this configuration. We follow the same convention for PatchTST and TimeMixer.

- PatchTST (Nie et al., 2023) has strongly influenced subsequent works by employing a global Transformer, in contrast to earlier local multivariate approaches that treated the group of input time series as a single multivariate series. PatchTST segments the time series and generates corresponding embeddings using an operation analogous to temporal convolution. Then, it applies attention over these segments, referred to as *patches*. It does not model spatial relations.

- TimeMixer (Wang et al., 2024b) is a fully MLP-based architecture that downsamples the input at different scales, decomposes it into trend and seasonal components, and employs feedforward layers to model temporal dependencies.

- Crossformer (Zhang and Yan, 2023) employs an input encoding with segmentation analogous to that used in PatchTST. The model is a hybrid global-local model, as it includes learnable position embeddings for each time series in the set. In addition to temporal attention, it captures spatial dependencies through attention over the spatial dimension using a routing mechanism. Furthermore, it adopts a hierarchical encoder-decoder structure.

- iTransformer (Liu et al., 2023a) is a global model that uses a feedforward approach to encode temporal dynamics and spatial attention to model spatial dependencies. This method has been described as applying attention to the *inverted dimension*, i.e., the spatial dimension. The model is global.

- ModernTCN (Donghao and Xue, 2024) uses depth-wise convolutions to encode temporal information, with an encoding similar to that performed in PatchTST, and then applies point-wise convolutions to process the feature and spatial dimensions separately.

- Linear (Toner and Darlow, 2024) is linear autoregressive models trained with $L2$ regularization and OLS. The *local* variant employs different weights for each series, while the *global* variant employs the same weights for all the series in the set.

## B  REFERENCE ARCHITECTURES STRUCTURE

The reference architectures, as schematized in Fig. 2, consist of a preprocessing module, followed by the processing of the temporal and spatial dynamics, and finally a postprocessing module. Their modular structure facilitates understanding of the architecture and promotes fair comparisons, as each module can be modified independently. In our experiments, we kept most modules fixed, modifying only the temporal and spatial modules for the experiments reported in Tables 3 and 11, respectively, and occasionally the feature encoding module. Here, we provide a more detailed description than the one given in Sec. 3.2.

**Preprocessing module**    The preprocessing module begins with RevInv (Kim et al., 2022) normalization. Then, the feature encoding moduleprocesses the input and covariates through non-linear layers and returns their sum. Alternatively, it can perform temporal convolution to generate an encoding similar to that in (Nie et al., 2023). Finally, local embeddings are concatenated with the resulting encoding.

**Processing module**    The processing module consists of temporal processing followed by spatial processing. In a more general architecture, these components could be interleaved. For simplicity, however, they are treated separately in our implementation.

**Postprocessing module**    The postprocessing module consists of a linear decoder that maps the hidden representations to predictions for the horizon. Finally, the predictions are de-normalized using the RevInv module.

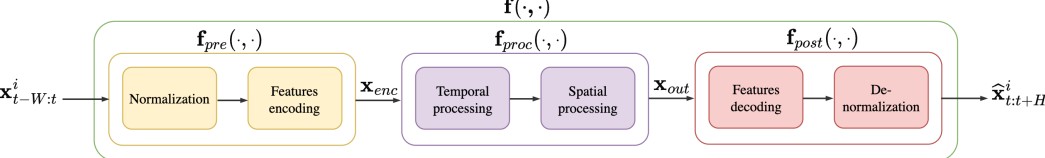

Figure 2: Block diagram of the reference architectures

## C EMPIRICAL SETUP AND ADDITIONAL EXPERIMENTAL RESULTS

Our code is implemented in Python (Van Rossum and Drake, 2009), with the use of the following libraries: PyTorch (Paszke et al., 2019); PyTorch Geometric (Fey and Lenssen, 2019); Torch Spatiotemporal (Cini and Marisca, 2022); Scikit-learn (Pedregosa et al., 2011); PyTorch Lightning (Falcon and The PyTorch Lightning team, 2019); Hydra (Yadan, 2019); Numpy (Harris et al., 2020). Below, we provide further details on the experiments conducted in Sec. 4 and report complete tables for both the MSE and mean absolute error (MAE) test metrics. Moreover, we provide additional information on the datasets in Tab. 6.

Table 6: Information on the datasets.

| Dataset | Time series | Steps | Frequency | Domain |
|---|---|---|---|---|
| Weather | 21 | 52695 | 10min | Weather |
| Solar-Energy | 137 | 52559 | 10min | Energy |
| ECL | 321 | 26303 | Hourly | Electricity |
| Traffic | 862 | 17543 | Hourly | Transportation |

**Hyperparameter tuning** For each experiment, we set a fixed batch size for each dataset. The hidden size is tuned between 32 and 256 for all datasets, with the addition of 16 for the Weather dataset. In Tab. 3, 12, 11, we used hyperparameters corresponding to the best configuration found during tuning. For Tab. 9, 10, Fig. 3, and 4, we used the same hyperparameters obtained from the tuning performed for Tab. 3, 11. Instead, in Tab. 7 and 8, we used fixed hyperparameters, identical for both sides of the comparison, without any tuning. The window size was set to 336 in Tab. 3, 9, and Fig. 3, except for the Solar dataset, for which it was set to 96. For the other tables, the window size was set to 96. Where not otherwise specified, the forecasting horizon is set to 96. Hyperparameters are fixed across horizons.

**Empirical setup for D1: model configuration** In Tab. 7, the global TimeMixer model is obtained by removing the learnable local parameters from the normalization module, while the global versions of both Crossformer and the reference Transformer are obtained by excluding their local embeddings. Analogously, the hybrid iTransformer is obtained by incorporating local embeddings into the input encoding.

**Empirical setup for D2: preprocessing and exogenous variables** In Tab. 8, covariates were removed from the reference Transformer and iTransformer, whereas they were included in the original implementations of PatchTST, DLinear, and Crossformer.

**Empirical setup for D3: temporal processing** In Tab. 3, we report the MSE and MAE performance of the reference architectures for various temporal processing modules, evaluated against baselines that do not include spatial processing operators. Tab. 9 presents the computational efficiency of the models for increasing horizons on the Electricity dataset. We employed the PyTorch Profiler (Paszke et al., 2019) to monitor GPU performance during training, specifically collecting the total CUDA execution time. Additionally, GPU memory usage was obtained using the PyG (Fey et al., 2025) function *get_gpu_memory_from_nvidia_smi*. To ensure a consistent evaluation, all measurements

Table 7: Comparison (MSE and MAE) of models with and without local parameters. Best average results are in **bold**.

| D | Model | hybrid | | global | |
|---|---|---|---|---|---|
| | | MSE | MAE | MSE | MAE |
| Electr. | Transf. | **0.136**±.000 | **0.231**±.000 | 0.151±.000 | 0.242±.000 |
| | Crossformer | **0.141**±.001 | **0.235**±.001 | 0.146±.003 | 0.240±.003 |
| | TimeMixer | **0.151**±.000 | **0.248**±.001 | 0.180±.001 | 0.268±.001 |
| | iTransformer | **0.139**±.000 | **0.236**±.001 | 0.154±.000 | 0.245±.000 |
| Weather | Transf. | **0.153**±.001 | **0.198**±.000 | 0.177±.002 | 0.215±.001 |
| | Crossformer | **0.154**±.003 | 0.225±.004 | 0.164±.003 | **0.224**±.007 |
| | TimeMixer | **0.164**±.002 | **0.208**±.001 | 0.178±.001 | 0.216±.001 |
| | iTransformer | **0.154**±.000 | **0.199**±.001 | 0.170±.001 | 0.211±.001 |
| Traffic | Transf. | 0.417±.009 | 0.278±.005 | **0.392**±.000 | **0.260**±.001 |
| | Crossformer | 0.540±.014 | 0.279±.007 | **0.512**±.007 | **0.259**±.004 |
| | TimeMixer | 0.464±.001 | 0.328±.003 | **0.463**±.001 | **0.327**±.003 |
| | iTransformer | 0.435±.002 | **0.275**±.000 | **0.409**±.000 | 0.277±.001 |
| Solar | Transf. | **0.196**±.000 | **0.243**±.001 | 0.205±.001 | 0.247±.002 |
| | Crossformer | 0.177±.008 | 0.215±.004 | **0.166**±.005 | **0.204**±.006 |
| | TimeMixer | **0.366**±.017 | **0.396**±.013 | 0.367±.017 | 0.396±.013 |
| | iTransformer | **0.189**±.001 | **0.240**±.004 | 0.197±.002 | 0.243±.001 |

Table 8: Comparison (MSE and MAE) of models with and without covariates. Best average results are in **bold**.

| D | Model | w/ exog. | | w/o exog. | |
|---|---|---|---|---|---|
| | | MSE | MAE | MSE | MAE |
| Electr. | Transf. | **0.136**±.000 | **0.231**±.000 | 0.155±.001 | 0.247±.000 |
| | PatchTST | **0.128**±.000 | **0.222**±.000 | 0.134±.000 | 0.228±.001 |
| | Crossformer | **0.139**±.002 | **0.234**±.003 | 0.141±.001 | 0.235±.001 |
| | iTransformer | **0.154**±.000 | **0.245**±.000 | 0.167±.000 | 0.254±.000 |
| | DLinear | **0.193**±.000 | 0.277±.000 | 0.195±.000 | 0.277±.000 |
| Weather | Transf. | **0.153**±.001 | **0.198**±.000 | 0.161±.000 | 0.208±.001 |
| | PatchTST | **0.174**±.000 | **0.213**±.001 | 0.180±.002 | 0.221±.002 |
| | Crossformer | 0.154±.003 | 0.225±.002 | 0.154±.003 | 0.225±.004 |
| | iTransformer | **0.170**±.001 | **0.211**±.001 | 0.176±.000 | 0.217±.001 |
| | DLinear | 0.199±.005 | 0.258±.008 | **0.196**±.001 | **0.248**±.002 |
| Traffic | Transf. | **0.417**±.009 | **0.278**±.005 | 0.479±.006 | 0.289±.001 |
| | PatchTST | **0.355**±.000 | **0.244**±.000 | 0.383±.001 | 0.261±.001 |
| | Crossformer | 0.548±.024 | **0.278**±.011 | **0.540**±.014 | 0.279±.007 |
| | iTransformer | **0.409**±.000 | **0.277**±.001 | 0.444±.001 | 0.290±.001 |
| | DLinear | **0.609**±.000 | **0.391**±.000 | 0.648±.000 | 0.395±.000 |
| Solar | Transf. | **0.196**±.000 | **0.243**±.001 | 0.206±.003 | 0.249±.004 |
| | PatchTST | **0.196**±.001 | **0.246**±.004 | 0.225±.003 | 0.268±.003 |
| | Crossformer | 0.176±.006 | 0.231±.010 | 0.177±.008 | **0.215**±.004 |
| | iTransformer | **0.197**±.002 | **0.243**±.001 | 0.221±.003 | 0.256±.002 |
| | DLinear | **0.246**±.001 | **0.331**±.000 | 0.285±.001 | 0.372±.001 |

related to model performance (Tab. 9 and 10) were conducted on the same machine running Oracle Linux Server 8.8, equipped with an Intel Xeon E5-2650 v3 CPU @ 2.30 GHz 20 (2 x 10) cores, 128 GB of system RAM, and an NVIDIA A100-PCIe GPU with 40 GB of HBM2 memory. Finally, we summarize these results in Fig. 3 which illustrate the trade-off between model performance and computational efficiency in terms of training batch time and GPU memory usage, on the Electricity dataset for a forecasting horizon of 96.

Table 9: Performance and resource utilization of the models selected in 3 on the Electricity dataset. Best performance is shown in **bold**, second best is underlined.

| Model | Horizon | Batch Time (ms) | Batches per Second | GPU Mem. (MB) | CUDA Time (ms) |
|---|---|---|---|---|---|
| MLP | 96 | $27.6_{\pm 1.4}$ | $36.3_{\pm 1.2}$ | 628.0 | 20.3 |
| | 192 | $27.6_{\pm 1.4}$ | $36.3_{\pm 1.2}$ | 628.0 | 27.1 |
| | 336 | $27.6_{\pm 1.4}$ | $36.3_{\pm 1.2}$ | 653.2 | 26.3 |
| | 720 | $27.6_{\pm 1.4}$ | $36.3_{\pm 1.2}$ | 705.6 | 31.8 |
| RNN | 96 | $36.7_{\pm 0.9}$ | $28.2_{\pm 0.6}$ | 3635.2 | 235.6 |
| | 192 | $37.3_{\pm 2.9}$ | $27.8_{\pm 1.8}$ | 3643.5 | 243.0 |
| | 336 | $37.3_{\pm 2.9}$ | $27.8_{\pm 1.8}$ | 3854.3 | 246.7 |
| | 720 | $37.3_{\pm 2.9}$ | $27.8_{\pm 1.8}$ | 3860.6 | 258.5 |
| TCN | 96 | $29.9_{\pm 1.4}$ | $33.5_{\pm 1.1}$ | 1217.3 | 62.0 |
| | 192 | $29.9_{\pm 1.4}$ | $33.5_{\pm 1.1}$ | 1217.3 | 82.0 |
| | 336 | $29.9_{\pm 1.4}$ | $33.5_{\pm 1.1}$ | 1219.4 | 85.1 |
| | 720 | $29.9_{\pm 1.4}$ | $33.5_{\pm 1.1}$ | 1240.4 | 87.8 |
| Transf. | 96 | $36.7_{\pm 1.3}$ | $27.3_{\pm 0.7}$ | 1942.9 | 119.3 |
| | 192 | $36.7_{\pm 1.3}$ | $27.3_{\pm 0.7}$ | 1961.7 | 144.0 |
| | 336 | $36.7_{\pm 1.3}$ | $27.3_{\pm 0.7}$ | 1959.7 | 148.1 |
| | 720 | $36.7_{\pm 1.3}$ | $27.3_{\pm 0.7}$ | 1976.4 | 164.7 |
| Pyraf. | 96 | $41.7_{\pm 0.8}$ | $24.0_{\pm 0.4}$ | 2559.4 | 189.9 |
| | 192 | $41.7_{\pm 0.8}$ | $24.0_{\pm 0.4}$ | 2561.5 | 191.9 |
| | 336 | $41.7_{\pm 0.8}$ | $24.0_{\pm 0.4}$ | 2563.6 | 192.8 |
| | 720 | $41.7_{\pm 0.8}$ | $24.0_{\pm 0.4}$ | 2567.8 | 198.7 |
| DLinear | 96 | $\mathbf{18.9_{\pm 1.1}}$ | $\mathbf{52.9_{\pm 1.7}}$ | **615.5** | **10.7** |
| | 192 | $\mathbf{18.9_{\pm 1.1}}$ | $\mathbf{52.9_{\pm 1.7}}$ | **615.5** | **17.2** |
| | 336 | $\mathbf{18.9_{\pm 1.1}}$ | $\mathbf{52.9_{\pm 1.7}}$ | **638.5** | **19.1** |
| | 720 | $\mathbf{18.9_{\pm 1.1}}$ | $\mathbf{52.9_{\pm 1.7}}$ | **699.4** | **22.5** |
| PatchTST | 96 | $34.3_{\pm 0.5}$ | $29.1_{\pm 0.4}$ | 1445.9 | 74.9 |
| | 192 | $34.3_{\pm 0.5}$ | $29.1_{\pm 0.4}$ | 1443.8 | 94.6 |
| | 336 | $34.3_{\pm 0.5}$ | $29.1_{\pm 0.4}$ | 1443.8 | 93.6 |
| | 720 | $34.3_{\pm 0.5}$ | $29.1_{\pm 0.4}$ | 1462.7 | 107.7 |
| TimeMixer | 96 | $97.6_{\pm 93}.5$ | $10.9_{\pm 0.7}$ | 2301.5 | 410.4 |
| | 192 | $97.6_{\pm 93}.5$ | $10.9_{\pm 0.7}$ | 2303.6 | 405.1 |
| | 336 | $97.6_{\pm 93}.5$ | $10.9_{\pm 0.7}$ | 2311.9 | 375.9 |
| | 720 | $97.6_{\pm 93}.5$ | $10.9_{\pm 0.7}$ | 2450.3 | 478.8 |

**Empirical setup for D4: spatial processing** In Tab. 11, we compare the reference architectures with baselines that include spatial processing. The reference architectures are employed with either an MLP or a pyramidal attention module for temporal processing, followed by a spatial attention module. The temporal modules were chosen for their advantageous trade-off between performance and computational efficiency (see Fig. 3 and Tab. 9). In Tab. 12, the iTransformer version without attention was obtained by replacing the spatial attention with a simple feedforward layer. Similarly to Fig. 3, Fig. 4 shows the trade-off between model performance and computational efficiency on the Electricity dataset for a forecasting horizon of 96. Since spatial processing often increases computational cost and reduces memory efficiency, we restrict the input window size to 96, equal to the forecasting horizon used in Tab. 11, 10, 12 and Fig. 4.

Table 10: Performance and resource utilization of the models selected in 4a on the Electricity dataset. Best performance is shown in **bold**, second best is underlined.

| Model | Horizon | Batch Time (ms) | Batches per Second | GPU Mem. (MB) | CUDA Time (ms) |
|---|---|---|---|---|---|
| MLP + sp. att. | 96 | **$40.2_{\pm1.0}$** | **$24.9_{\pm0.5}$** | **1162.8** | **96.9** |
| | 192 | **$40.6_{\pm1.3}$** | **$24.7_{\pm0.6}$** | **1236.2** | **123.8** |
| | 336 | **$40.6_{\pm1.3}$** | **$24.7_{\pm0.6}$** | **1215.2** | **155.4** |
| | 720 | **$40.6_{\pm1.3}$** | **$24.7_{\pm0.6}$** | **1357.8** | **185.8** |
| Pyraf. + sp. att. | 96 | $96.4_{\pm0.4}$ | $10.4_{\pm0.0}$ | 7822.9 | 783.6 |
| | 192 | $96.4_{\pm0.4}$ | $10.4_{\pm0.0}$ | 7908.8 | 787.3 |
| | 336 | $96.4_{\pm0.4}$ | $10.4_{\pm0.0}$ | 7837.5 | 808.8 |
| | 720 | $96.4_{\pm0.4}$ | $10.4_{\pm0.0}$ | 7940.3 | 861.9 |
| iTransformer | 96 | $50.8_{\pm1.3}$ | $19.7_{\pm0.4}$ | 1729.0 | 217.4 |
| | 192 | $50.7_{\pm1.1}$ | $19.7_{\pm0.4}$ | 1630.4 | 227.9 |
| | 336 | $50.7_{\pm1.1}$ | $19.7_{\pm0.4}$ | 1712.2 | 252.9 |
| | 720 | $50.7_{\pm1.1}$ | $19.7_{\pm0.4}$ | 1871.6 | 317.8 |
| Crossformer | 96 | $138.0_{\pm1.0}$ | $7.2_{\pm0.1}$ | 5702.8 | 912.8 |
| | 192 | $161.9_{\pm28.1}$ | $6.4_{\pm1.0}$ | 9412.4 | 1532.0 |
| | 336 | $161.9_{\pm28.1}$ | $6.4_{\pm1.0}$ | 16032.6 | 2516.0 |
| | 720 | $161.9_{\pm28.1}$ | $6.4_{\pm1.0}$ | 39527.4 | 5589.0 |
| Modern TCN | 96 | $145.4_{\pm1.1}$ | $6.9_{\pm0.0}$ | 10620.3 | 1967.0 |
| | 192 | $147.4_{\pm8.8}$ | $6.8_{\pm0.3}$ | 10620.3 | 1971.0 |
| | 336 | $147.4_{\pm8.8}$ | $6.8_{\pm0.3}$ | 10662.2 | 2005.0 |
| | 720 | $147.4_{\pm8.8}$ | $6.8_{\pm0.3}$ | 10857.2 | 2036.0 |

Table 11: Forecasting results (MSE and MAE) for a horizon of 96 steps for models *including* spatial processing. Best average results are in **bold**, second best are underlined.

| Model | Electricity | | Weather | | Traffic | | Solar | |
|---|---|---|---|---|---|---|---|---|
| | MSE | MAE | MSE | MAE | MSE | MAE | MSE | MAE |
| MLP + sp. attn. | $0.140_{\pm.001}$ | $0.238_{\pm.001}$ | $0.157_{\pm.000}$ | $0.202_{\pm.001}$ | $0.435_{\pm.006}$ | $0.275_{\pm.001}$ | $0.201_{\pm.009}$ | $0.246_{\pm.003}$ |
| Pyraf. + sp. attn. | $0.139_{\pm.001}$ | $0.236_{\pm.001}$ | $0.157_{\pm.002}$ | $0.204_{\pm.001}$ | **$0.389_{\pm.002}$** | $0.267_{\pm.001}$ | $0.188_{\pm.002}$ | $0.235_{\pm.003}$ |
| iTransformer | $0.148_{\pm.000}$ | $0.241_{\pm.000}$ | $0.171_{\pm.001}$ | $0.210_{\pm.001}$ | $0.393_{\pm.001}$ | **$0.266_{\pm.001}$** | $0.208_{\pm.003}$ | $0.240_{\pm.006}$ |
| Crosformer | **$0.136_{\pm.000}$** | **$0.232_{\pm.001}$** | **$0.152_{\pm.003}$** | $0.222_{\pm.004}$ | $0.527_{\pm.002}$ | $0.270_{\pm.003}$ | **$0.184_{\pm.008}$** | $0.227_{\pm.006}$ |
| ModernTCN | $0.141_{\pm.000}$ | $0.237_{\pm.001}$ | $0.154_{\pm.001}$ | **$0.200_{\pm.001}$** | $0.445_{\pm.001}$ | $0.287_{\pm.001}$ | $0.190_{\pm.001}$ | **$0.222_{\pm.002}$** |

Table 12: Results (MSE and MAE) for iTransformer with or without space attention. Best average results are in **bold**.

| Dataset | Space att. | | Feedforward | |
|---|---|---|---|---|
| | MSE | MAE | MSE | MAE |
| Electricity | **$0.148_{\pm.000}$** | $0.241_{\pm.000}$ | $0.149_{\pm.001}$ | **$0.237_{\pm.001}$** |
| Weather | $0.171_{\pm.001}$ | $0.210_{\pm.001}$ | $0.171_{\pm.000}$ | $0.210_{\pm.001}$ |
| Traffic | $0.393_{\pm.001}$ | $0.266_{\pm.001}$ | **$0.390_{\pm.001}$** | **$0.258_{\pm.000}$** |
| Solar | $0.208_{\pm.003}$ | $0.240_{\pm.006}$ | **$0.194_{\pm.001}$** | **$0.230_{\pm.002}$** |

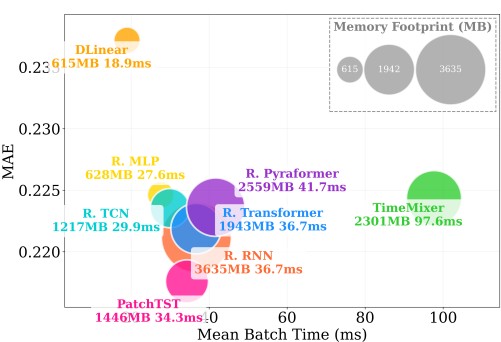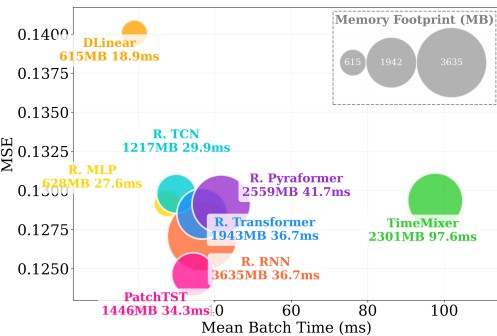

Figure 3: MAE and MSE performance versus mean batch time during training for models *not including* spatial processing, for a batch size of 512. Circle size indicates memory consumption.

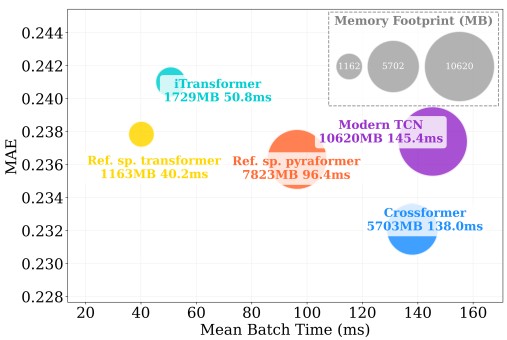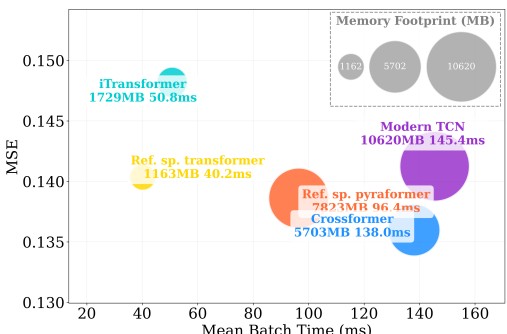

Figure 4: MAE and MSE performance versus mean batch time during training for models *including* spatial processing, for a batch size of 32. Circle size indicates memory consumption.

## D  MODEL CARDS

In Tab. 13, we report an example of the usage of the newly introduced model cards for PatchTST.

## E  ADDITIONAL RESULTS

We present additional results covering a wider range of input windows, forecasting horizons, and model configurations.

**Multi-horizon results**  Tab. 14 and Tab. 20 extend the results of Sec. 4.3 and Sec. 4.4 to a broader set of horizons (96, 192, 336, 720). We observe that increasing the forecasting horizon does not change the conclusions drawn in the corresponding sections, and results still show that—even for larger forecasting horizons—standard, streamlined architectures achieve performance comparable to current state-of-the-art models.

**Extended ablation on iTransformer**  Tab. 15, Tab. 16, Tab. 19 and Tab. 17 expand the iTransformer ablation study of Tab. 12 to additional window sizes (96, 336, 720) and forecasting horizons (96, 192, 336, 720). These additional experiments further reinforce our analysis in Sec. 4.4, showing a general performance improvement when removing one of the core components introduced by state-of-the-art architectures.

**Linear autoregressive models**  In Tab. 18, we evaluate the performance of two simple hybrid variants of Linear and DLinear against the local and global configurations. The hybrid DLinear variant is obtained by concatenating learnable local parameters to the input, whereas for Linear, we concatenate a one-hot encoding representing the time series. The results are consistent with the findings discussed in Sec. 4.1.

Table 13: Example of model cards for PatchTST on the Electricity dataset

---

**FORECASTING MODEL CARD**

**Model setting**

- *Window length*: fixed lookback window of 336
- *Transductive or inductive (cold start)*: inductive
- *Masking*: not applied/needed

**D1. Model configuration**

- *Global/local/hybrid*: global model
- *Hybrid parameters (non-shared)*: not applicable

**D2. Preprocessing and exogenous variables**

- *Scaling*: standard normalization (z-score) applied per series and in-batch RevInv normalization
- *Covariates/exogenous variables*: not used

**D3. Temporal processing**

- *Temporal modules*: convolutional encoding followed by patching-based Transformer layers
- *Complexity scaling with steps*: the time and space complexity scales quadratically with the number of patches (self-attention)

**D4. Spatial processing**

- *Spatial modules*: not applicable
- *Complexity scaling with nodes*: not applicable

---

## F  LARGE LANGUAGE MODELS

We acknowledge the use of Large Language Models to assist in polishing the manuscript by making minor edits to single sentences.

## G  CODE OF ETHICS

The work presented in this paper is about basic machine learning research, and all experiments are conducted on standard, publicly available datasets. The authors have read and adhere to the ICLR Code of Ethics and do not foresee any direct ethical concerns or potential for misuse.

Table 14: Results (MSE and MAE) for multiple horizons. Best mean results are in **bold**, second best are underlined.

| Model | H | Electricity | | Weather | | Traffic | | Solar | |
|---|---|---|---|---|---|---|---|---|---|
| | | MSE | MAE | MSE | MAE | MSE | MAE | MSE | MAE |
| OLS Global | 96 | 0.140 | 0.237 | 0.174 | 0.234 | 0.410 | 0.282 | 0.222 | 0.291 |
| | 192 | 0.154 | 0.250 | 0.215 | 0.272 | 0.423 | 0.288 | 0.249 | 0.309 |
| | 336 | 0.169 | 0.268 | 0.260 | 0.309 | 0.436 | 0.295 | 0.269 | 0.324 |
| | 720 | 0.204 | 0.301 | 0.323 | 0.361 | 0.466 | 0.315 | 0.271 | 0.327 |
| OLS Local | 96 | 0.134 | 0.230 | **0.144** | 0.209 | 0.426 | 0.298 | 0.223 | 0.295 |
| | 192 | 0.149 | 0.245 | **0.187** | 0.254 | 0.438 | 0.304 | 0.251 | 0.313 |
| | 336 | 0.165 | 0.263 | **0.240** | 0.298 | 0.452 | 0.312 | 0.270 | 0.327 |
| | 720 | 0.201 | 0.297 | **0.316** | 0.358 | 0.482 | 0.330 | 0.272 | 0.331 |
| R. MLP | 96 | $0.129_{\pm0.000}$ | $0.225_{\pm0.000}$ | $0.148_{\pm0.001}$ | $0.198_{\pm0.000}$ | $0.376_{\pm0.000}$ | $0.253_{\pm0.001}$ | $0.194_{\pm0.003}$ | $0.239_{\pm0.002}$ |
| | 192 | $0.149_{\pm0.000}$ | $0.245_{\pm0.001}$ | $0.191_{\pm0.001}$ | $0.241_{\pm0.000}$ | $0.403_{\pm0.001}$ | $0.270_{\pm0.001}$ | $0.227_{\pm0.001}$ | $0.262_{\pm0.001}$ |
| | 336 | $0.166_{\pm0.000}$ | $0.262_{\pm0.000}$ | $0.245_{\pm0.001}$ | $0.281_{\pm0.001}$ | $0.417_{\pm0.001}$ | $0.277_{\pm0.001}$ | $0.249_{\pm0.001}$ | $0.277_{\pm0.002}$ |
| | 720 | $0.205_{\pm0.000}$ | $0.296_{\pm0.000}$ | $0.324_{\pm0.002}$ | $0.338_{\pm0.001}$ | $0.456_{\pm0.001}$ | $0.295_{\pm0.001}$ | $0.254_{\pm0.000}$ | $0.279_{\pm0.001}$ |
| R. RNN | 96 | $0.127_{\pm0.001}$ | $0.221_{\pm0.000}$ | $0.148_{\pm0.000}$ | $0.199_{\pm0.001}$ | $0.362_{\pm0.002}$ | $0.248_{\pm0.001}$ | $0.192_{\pm0.002}$ | **$0.236_{\pm0.001}$** |
| | 192 | $0.167_{\pm0.001}$ | $0.267_{\pm0.001}$ | $0.190_{\pm0.001}$ | $0.244_{\pm0.001}$ | $0.411_{\pm0.005}$ | $0.282_{\pm0.005}$ | $0.230_{\pm0.003}$ | $0.270_{\pm0.002}$ |
| | 336 | $0.186_{\pm0.001}$ | $0.287_{\pm0.001}$ | $0.244_{\pm0.001}$ | $0.286_{\pm0.001}$ | $0.423_{\pm0.019}$ | $0.285_{\pm0.005}$ | $0.251_{\pm0.007}$ | $0.284_{\pm0.007}$ |
| | 720 | $0.225_{\pm0.002}$ | $0.323_{\pm0.001}$ | $0.324_{\pm0.003}$ | $0.342_{\pm0.003}$ | $0.497_{\pm0.040}$ | $0.297_{\pm0.001}$ | $0.249_{\pm0.005}$ | $0.281_{\pm0.005}$ |
| R. TCN | 96 | $0.130_{\pm0.000}$ | $0.224_{\pm0.000}$ | $0.148_{\pm0.000}$ | $0.200_{\pm0.001}$ | $0.364_{\pm0.003}$ | $0.253_{\pm0.002}$ | $0.193_{\pm0.004}$ | $0.243_{\pm0.005}$ |
| | 192 | $0.148_{\pm0.000}$ | $0.240_{\pm0.000}$ | $0.195_{\pm0.001}$ | $0.246_{\pm0.000}$ | $0.382_{\pm0.001}$ | $0.261_{\pm0.001}$ | **$0.221_{\pm0.002}$** | **$0.253_{\pm0.002}$** |
| | 336 | $0.165_{\pm0.001}$ | $0.258_{\pm0.001}$ | $0.252_{\pm0.002}$ | $0.289_{\pm0.001}$ | $0.397_{\pm0.002}$ | $0.274_{\pm0.005}$ | $0.249_{\pm0.005}$ | **$0.271_{\pm0.005}$** |
| | 720 | $0.202_{\pm0.002}$ | $0.292_{\pm0.001}$ | $0.329_{\pm0.001}$ | $0.342_{\pm0.001}$ | **$0.434_{\pm0.001}$** | $0.291_{\pm0.005}$ | $0.246_{\pm0.004}$ | $0.273_{\pm0.003}$ |
| R. Transf. | 96 | $0.129_{\pm0.001}$ | $0.222_{\pm0.001}$ | $0.149_{\pm0.001}$ | $0.203_{\pm0.002}$ | $0.362_{\pm0.003}$ | $0.249_{\pm0.002}$ | $0.203_{\pm0.006}$ | $0.245_{\pm0.002}$ |
| | 192 | $0.146_{\pm0.000}$ | $0.238_{\pm0.000}$ | $0.198_{\pm0.001}$ | $0.250_{\pm0.002}$ | **$0.374_{\pm0.001}$** | **$0.255_{\pm0.001}$** | $0.224_{\pm0.001}$ | $0.260_{\pm0.001}$ |
| | 336 | $0.164_{\pm0.001}$ | $0.258_{\pm0.001}$ | $0.248_{\pm0.001}$ | $0.289_{\pm0.002}$ | **$0.393_{\pm0.002}$** | $0.271_{\pm0.007}$ | $0.245_{\pm0.004}$ | **$0.271_{\pm0.003}$** |
| | 720 | $0.202_{\pm0.003}$ | $0.294_{\pm0.002}$ | $0.330_{\pm0.010}$ | $0.344_{\pm0.006}$ | **$0.434_{\pm0.004}$** | $0.294_{\pm0.008}$ | $0.248_{\pm0.004}$ | $0.277_{\pm0.004}$ |
| R. Pyraf. | 96 | $0.129_{\pm0.001}$ | $0.224_{\pm0.001}$ | $0.148_{\pm0.001}$ | $0.199_{\pm0.001}$ | $0.365_{\pm0.002}$ | $0.251_{\pm0.003}$ | **$0.189_{\pm0.003}$** | **$0.236_{\pm0.004}$** |
| | 192 | $0.147_{\pm0.001}$ | $0.240_{\pm0.001}$ | $0.196_{\pm0.003}$ | $0.246_{\pm0.002}$ | $0.384_{\pm0.003}$ | $0.262_{\pm0.004}$ | $0.224_{\pm0.001}$ | $0.256_{\pm0.001}$ |
| | 336 | $0.164_{\pm0.001}$ | $0.258_{\pm0.000}$ | $0.248_{\pm0.003}$ | $0.287_{\pm0.003}$ | $0.397_{\pm0.001}$ | $0.269_{\pm0.002}$ | **$0.244_{\pm0.001}$** | **$0.271_{\pm0.002}$** |
| | 720 | $0.200_{\pm0.000}$ | $0.293_{\pm0.000}$ | $0.328_{\pm0.003}$ | $0.344_{\pm0.002}$ | **$0.434_{\pm0.002}$** | $0.297_{\pm0.003}$ | $0.248_{\pm0.002}$ | **$0.272_{\pm0.001}$** |
| TimeMixer | 96 | $0.129_{\pm0.001}$ | $0.224_{\pm0.000}$ | $0.147_{\pm0.001}$ | $0.197_{\pm0.000}$ | $0.373_{\pm0.002}$ | $0.271_{\pm0.003}$ | $0.199_{\pm0.001}$ | $0.245_{\pm0.000}$ |
| | 192 | $0.147_{\pm0.001}$ | $0.241_{\pm0.000}$ | $0.191_{\pm0.000}$ | **$0.239_{\pm0.000}$** | $0.396_{\pm0.001}$ | $0.283_{\pm0.001}$ | $0.230_{\pm0.003}$ | $0.268_{\pm0.003}$ |
| | 336 | $0.166_{\pm0.001}$ | $0.260_{\pm0.001}$ | $0.244_{\pm0.002}$ | $0.281_{\pm0.002}$ | $0.416_{\pm0.001}$ | $0.297_{\pm0.001}$ | **$0.244_{\pm0.002}$** | $0.280_{\pm0.000}$ |
| | 720 | $0.206_{\pm0.003}$ | $0.297_{\pm0.003}$ | $0.321_{\pm0.003}$ | $0.334_{\pm0.003}$ | $0.450_{\pm0.006}$ | $0.318_{\pm0.003}$ | **$0.245_{\pm0.004}$** | $0.280_{\pm0.001}$ |
| PatchTST | 96 | **$0.125_{\pm0.000}$** | **$0.218_{\pm0.000}$** | $0.148_{\pm0.001}$ | **$0.195_{\pm0.001}$** | **$0.345_{\pm0.000}$** | **$0.234_{\pm0.000}$** | $0.197_{\pm0.001}$ | $0.244_{\pm0.004}$ |
| | 192 | **$0.143_{\pm0.000}$** | **$0.236_{\pm0.000}$** | $0.194_{\pm0.000}$ | **$0.239_{\pm0.001}$** | $0.384_{\pm0.001}$ | $0.258_{\pm0.001}$ | $0.227_{\pm0.002}$ | $0.260_{\pm0.002}$ |
| | 336 | **$0.160_{\pm0.000}$** | **$0.254_{\pm0.000}$** | $0.247_{\pm0.000}$ | **$0.279_{\pm0.001}$** | $0.396_{\pm0.001}$ | **$0.264_{\pm0.000}$** | $0.249_{\pm0.001}$ | $0.273_{\pm0.002}$ |
| | 720 | **$0.197_{\pm0.001}$** | **$0.288_{\pm0.001}$** | $0.322_{\pm0.002}$ | **$0.333_{\pm0.001}$** | $0.435_{\pm0.001}$ | **$0.286_{\pm0.000}$** | $0.247_{\pm0.001}$ | $0.273_{\pm0.002}$ |
| DLinear | 96 | $0.140_{\pm0.000}$ | $0.237_{\pm0.000}$ | $0.173_{\pm0.000}$ | $0.232_{\pm0.001}$ | $0.407_{\pm0.000}$ | $0.283_{\pm0.000}$ | $0.246_{\pm0.001}$ | $0.331_{\pm0.000}$ |
| | 192 | $0.154_{\pm0.000}$ | $0.250_{\pm0.001}$ | $0.216_{\pm0.001}$ | $0.274_{\pm0.003}$ | $0.421_{\pm0.000}$ | $0.290_{\pm0.000}$ | $0.267_{\pm0.001}$ | $0.342_{\pm0.000}$ |
| | 336 | $0.169_{\pm0.000}$ | $0.268_{\pm0.000}$ | $0.265_{\pm0.002}$ | $0.318_{\pm0.003}$ | $0.433_{\pm0.000}$ | $0.296_{\pm0.000}$ | $0.289_{\pm0.001}$ | $0.353_{\pm0.000}$ |
| | 720 | $0.203_{\pm0.000}$ | $0.300_{\pm0.000}$ | $0.331_{\pm0.002}$ | $0.373_{\pm0.003}$ | $0.461_{\pm0.000}$ | $0.314_{\pm0.000}$ | $0.294_{\pm0.001}$ | $0.355_{\pm0.000}$ |

Table 15: Ablation study on itransformer for window=96 across different forecasting horizons. Best mean results are in **bold**.

| Dataset | Horizon | With space attention | | Without space attention | |
|---|---|---|---|---|---|
| | | MSE | MAE | MSE | MAE |
| Electricity | 96 | $\mathbf{0.148}_{\pm 0.000}$ | $0.241_{\pm 0.000}$ | $0.149_{\pm 0.001}$ | $\mathbf{0.237}_{\pm 0.001}$ |
| | 192 | $0.166_{\pm 0.001}$ | $0.259_{\pm 0.001}$ | $\mathbf{0.161}_{\pm 0.000}$ | $\mathbf{0.250}_{\pm 0.000}$ |
| | 336 | $0.179_{\pm 0.002}$ | $0.274_{\pm 0.002}$ | $0.179_{\pm 0.000}$ | $\mathbf{0.268}_{\pm 0.000}$ |
| | 720 | $\mathbf{0.218}_{\pm 0.000}$ | $0.309_{\pm 0.001}$ | $0.219_{\pm 0.000}$ | $\mathbf{0.303}_{\pm 0.000}$ |
| Weather | 96 | $0.171_{\pm 0.001}$ | $0.210_{\pm 0.001}$ | $0.171_{\pm 0.000}$ | $0.210_{\pm 0.001}$ |
| | 192 | $0.221_{\pm 0.001}$ | $0.255_{\pm 0.000}$ | $\mathbf{0.219}_{\pm 0.001}$ | $\mathbf{0.253}_{\pm 0.001}$ |
| | 336 | $0.276_{\pm 0.000}$ | $0.295_{\pm 0.000}$ | $\mathbf{0.275}_{\pm 0.000}$ | $\mathbf{0.294}_{\pm 0.000}$ |
| | 720 | $0.353_{\pm 0.000}$ | $0.346_{\pm 0.000}$ | $\mathbf{0.352}_{\pm 0.001}$ | $\mathbf{0.345}_{\pm 0.000}$ |
| Traffic | 96 | $0.393_{\pm 0.001}$ | $0.266_{\pm 0.001}$ | $\mathbf{0.390}_{\pm 0.001}$ | $\mathbf{0.258}_{\pm 0.000}$ |
| | 192 | $0.424_{\pm 0.001}$ | $0.286_{\pm 0.001}$ | $\mathbf{0.409}_{\pm 0.000}$ | $\mathbf{0.268}_{\pm 0.000}$ |
| | 336 | $0.430_{\pm 0.001}$ | $0.289_{\pm 0.001}$ | $\mathbf{0.423}_{\pm 0.000}$ | $\mathbf{0.274}_{\pm 0.000}$ |
| | 720 | $0.455_{\pm 0.001}$ | $0.303_{\pm 0.000}$ | $\mathbf{0.454}_{\pm 0.000}$ | $\mathbf{0.292}_{\pm 0.000}$ |
| Solar | 96 | $0.208_{\pm 0.003}$ | $0.240_{\pm 0.006}$ | $\mathbf{0.194}_{\pm 0.001}$ | $\mathbf{0.230}_{\pm 0.002}$ |
| | 192 | $0.233_{\pm 0.003}$ | $0.257_{\pm 0.003}$ | $\mathbf{0.226}_{\pm 0.001}$ | $0.257_{\pm 0.003}$ |
| | 336 | $\mathbf{0.244}_{\pm 0.000}$ | $0.273_{\pm 0.002}$ | $0.245_{\pm 0.003}$ | $\mathbf{0.267}_{\pm 0.002}$ |
| | 720 | $0.255_{\pm 0.002}$ | $0.280_{\pm 0.001}$ | $\mathbf{0.249}_{\pm 0.001}$ | $\mathbf{0.271}_{\pm 0.001}$ |

Table 16: Ablation study on itransformer for window=336 across different forecasting horizons. Best mean results are in **bold**.

| Dataset | Horizon | With space attention | | Without space attention | |
|---|---|---|---|---|---|
| | | MSE | MAE | MSE | MAE |
| Electricity | 96 | $0.135_{\pm 0.001}$ | $0.229_{\pm 0.001}$ | $\mathbf{0.130}_{\pm 0.000}$ | $\mathbf{0.223}_{\pm 0.000}$ |
| | 192 | $0.155_{\pm 0.001}$ | $0.248_{\pm 0.001}$ | $\mathbf{0.149}_{\pm 0.000}$ | $\mathbf{0.242}_{\pm 0.000}$ |
| | 336 | $0.172_{\pm 0.002}$ | $0.267_{\pm 0.000}$ | $\mathbf{0.166}_{\pm 0.000}$ | $\mathbf{0.260}_{\pm 0.000}$ |
| | 720 | $\mathbf{0.201}_{\pm 0.002}$ | $0.294_{\pm 0.002}$ | $0.205_{\pm 0.000}$ | $0.294_{\pm 0.000}$ |
| Weather | 96 | $0.159_{\pm 0.001}$ | $0.207_{\pm 0.000}$ | $\mathbf{0.153}_{\pm 0.000}$ | $\mathbf{0.202}_{\pm 0.001}$ |
| | 192 | $0.203_{\pm 0.001}$ | $0.249_{\pm 0.001}$ | $\mathbf{0.197}_{\pm 0.001}$ | $\mathbf{0.245}_{\pm 0.002}$ |
| | 336 | $0.252_{\pm 0.002}$ | $0.286_{\pm 0.000}$ | $\mathbf{0.249}_{\pm 0.001}$ | $\mathbf{0.284}_{\pm 0.001}$ |
| | 720 | $\mathbf{0.326}_{\pm 0.003}$ | $\mathbf{0.338}_{\pm 0.002}$ | $0.328_{\pm 0.002}$ | $0.339_{\pm 0.001}$ |
| Traffic | 96 | $0.363_{\pm 0.000}$ | $0.257_{\pm 0.001}$ | $\mathbf{0.359}_{\pm 0.001}$ | $\mathbf{0.247}_{\pm 0.001}$ |
| | 192 | $0.385_{\pm 0.002}$ | $0.269_{\pm 0.001}$ | $\mathbf{0.377}_{\pm 0.001}$ | $\mathbf{0.256}_{\pm 0.001}$ |
| | 336 | $0.397_{\pm 0.002}$ | $0.277_{\pm 0.001}$ | $\mathbf{0.386}_{\pm 0.000}$ | $\mathbf{0.261}_{\pm 0.000}$ |
| | 720 | $0.423_{\pm 0.001}$ | $0.291_{\pm 0.000}$ | $\mathbf{0.420}_{\pm 0.001}$ | $\mathbf{0.282}_{\pm 0.000}$ |
| Solar | 96 | $0.195_{\pm 0.000}$ | $0.252_{\pm 0.003}$ | $\mathbf{0.189}_{\pm 0.001}$ | $\mathbf{0.232}_{\pm 0.001}$ |
| | 192 | $0.222_{\pm 0.004}$ | $0.269_{\pm 0.001}$ | $\mathbf{0.207}_{\pm 0.000}$ | $\mathbf{0.249}_{\pm 0.000}$ |
| | 336 | $0.230_{\pm 0.007}$ | $0.276_{\pm 0.004}$ | $\mathbf{0.214}_{\pm 0.000}$ | $\mathbf{0.255}_{\pm 0.001}$ |
| | 720 | $0.223_{\pm 0.002}$ | $0.274_{\pm 0.003}$ | $\mathbf{0.216}_{\pm 0.003}$ | $\mathbf{0.258}_{\pm 0.002}$ |

Table 17: Ablation study on itransformer for horizon=96 across different window lengths. Best mean results are in **bold**.

| Dataset | Window | With space attention | | Without space attention | |
|---|---|---|---|---|---|
| | | MSE | MAE | MSE | MAE |
| Electricity | 96 | $\mathbf{0.148}_{\pm 0.000}$ | $0.241_{\pm 0.000}$ | $0.149_{\pm 0.001}$ | $\mathbf{0.237}_{\pm 0.001}$ |
| | 336 | $0.135_{\pm 0.001}$ | $0.229_{\pm 0.001}$ | $\mathbf{0.130}_{\pm 0.000}$ | $\mathbf{0.223}_{\pm 0.000}$ |
| | 720 | $0.135_{\pm 0.002}$ | $0.231_{\pm 0.001}$ | $\mathbf{0.132}_{\pm 0.000}$ | $\mathbf{0.227}_{\pm 0.001}$ |
| Weather | 96 | $0.171_{\pm 0.001}$ | $0.210_{\pm 0.001}$ | $0.171_{\pm 0.000}$ | $0.210_{\pm 0.001}$ |
| | 336 | $0.159_{\pm 0.001}$ | $0.207_{\pm 0.000}$ | $\mathbf{0.153}_{\pm 0.000}$ | $\mathbf{0.202}_{\pm 0.001}$ |
| | 720 | $0.155_{\pm 0.002}$ | $0.208_{\pm 0.002}$ | $\mathbf{0.149}_{\pm 0.000}$ | $\mathbf{0.201}_{\pm 0.001}$ |
| Traffic | 96 | $0.393_{\pm 0.001}$ | $0.266_{\pm 0.001}$ | $\mathbf{0.390}_{\pm 0.001}$ | $\mathbf{0.258}_{\pm 0.000}$ |
| | 336 | $0.363_{\pm 0.000}$ | $0.257_{\pm 0.001}$ | $\mathbf{0.359}_{\pm 0.001}$ | $\mathbf{0.247}_{\pm 0.001}$ |
| | 720 | $\mathbf{0.353}_{\pm 0.004}$ | $0.256_{\pm 0.001}$ | $0.357_{\pm 0.000}$ | $\mathbf{0.251}_{\pm 0.000}$ |
| Solar | 96 | $0.208_{\pm 0.003}$ | $0.240_{\pm 0.006}$ | $\mathbf{0.194}_{\pm 0.001}$ | $\mathbf{0.230}_{\pm 0.002}$ |
| | 336 | $0.195_{\pm 0.000}$ | $0.252_{\pm 0.003}$ | $\mathbf{0.189}_{\pm 0.001}$ | $\mathbf{0.232}_{\pm 0.001}$ |
| | 720 | $\mathbf{0.175}_{\pm 0.003}$ | $0.245_{\pm 0.004}$ | $0.182_{\pm 0.008}$ | $\mathbf{0.235}_{\pm 0.006}$ |

Table 18: Comparison (MSE and MAE) of Linear models in their local, global, and hybrid variants. A dash (–) marks experiments beyond our computational budget. Best mean results are in **bold**.

| Dataset | Model | Hybrid | | Global | | Local | |
|---|---|---|---|---|---|---|---|
| | | MSE | MAE | MSE | MAE | MSE | MAE |
| Electr. | DLinear | $0.195_{\pm .000}$ | $0.277_{\pm .000}$ | $0.195_{\pm .000}$ | $0.277_{\pm .000}$ | $\mathbf{0.184}_{\pm .000}$ | $\mathbf{0.270}_{\pm .000}$ |
| | OLS | $0.194_{\pm .000}$ | $0.277_{\pm .000}$ | $0.195_{\pm .000}$ | $0.277_{\pm .000}$ | $\mathbf{0.184}_{\pm .000}$ | $\mathbf{0.270}_{\pm .000}$ |
| Weather | DLinear | $0.198_{\pm .001}$ | $0.254_{\pm .002}$ | $0.196_{\pm .001}$ | $0.248_{\pm .002}$ | $\mathbf{0.161}_{\pm .001}$ | $\mathbf{0.233}_{\pm .001}$ |
| | OLS | $0.196_{\pm .000}$ | $0.254_{\pm .000}$ | $0.195_{\pm .000}$ | $0.253_{\pm .000}$ | $\mathbf{0.161}_{\pm .000}$ | $\mathbf{0.233}_{\pm .000}$ |
| Traffic | DLinear | $0.648_{\pm .000}$ | $\mathbf{0.395}_{\pm .000}$ | $0.648_{\pm .000}$ | $\mathbf{0.395}_{\pm .000}$ | $0.647_{\pm .000}$ | $0.403_{\pm .000}$ |
| | OLS | - | - | $0.649_{\pm .000}$ | $\mathbf{0.396}_{\pm .000}$ | $\mathbf{0.647}_{\pm .000}$ | $0.403_{\pm .000}$ |
| Solar | DLinear | $0.286_{\pm .000}$ | $0.375_{\pm .001}$ | $\mathbf{0.285}_{\pm .001}$ | $\mathbf{0.372}_{\pm .001}$ | $0.286_{\pm .000}$ | $0.376_{\pm .001}$ |
| | OLS | $\mathbf{0.285}_{\pm .000}$ | $\mathbf{0.372}_{\pm .000}$ | $\mathbf{0.285}_{\pm .000}$ | $\mathbf{0.372}_{\pm .000}$ | $0.286_{\pm .000}$ | $0.374_{\pm .000}$ |

Table 19: Ablation study on itransformer for window=720 across different forecasting horizons. Best mean results are in **bold**.

| Dataset | Horizon | With space attention | | Without space attention | |
|---|---|---|---|---|---|
| | | MSE | MAE | MSE | MAE |
| Electricity | 96 | $0.135_{\pm0.002}$ | $0.231_{\pm0.001}$ | $\mathbf{0.132_{\pm0.000}}$ | $\mathbf{0.227_{\pm0.001}}$ |
| | 192 | $0.157_{\pm0.001}$ | $0.252_{\pm0.001}$ | $\mathbf{0.151_{\pm0.001}}$ | $\mathbf{0.245_{\pm0.001}}$ |
| | 336 | $0.175_{\pm0.001}$ | $0.272_{\pm0.001}$ | $\mathbf{0.165_{\pm0.000}}$ | $\mathbf{0.263_{\pm0.000}}$ |
| | 720 | $\mathbf{0.195_{\pm0.001}}$ | $\mathbf{0.289_{\pm0.002}}$ | $0.201_{\pm0.001}$ | $0.294_{\pm0.001}$ |
| Weather | 96 | $0.155_{\pm0.002}$ | $0.208_{\pm0.002}$ | $\mathbf{0.149_{\pm0.000}}$ | $\mathbf{0.201_{\pm0.001}}$ |
| | 192 | $0.202_{\pm0.002}$ | $0.250_{\pm0.001}$ | $\mathbf{0.195_{\pm0.001}}$ | $\mathbf{0.247_{\pm0.002}}$ |
| | 336 | $0.249_{\pm0.002}$ | $\mathbf{0.288_{\pm0.002}}$ | $0.249_{\pm0.001}$ | $0.289_{\pm0.001}$ |
| | 720 | $0.322_{\pm0.002}$ | $0.342_{\pm0.000}$ | $\mathbf{0.317_{\pm0.001}}$ | $\mathbf{0.337_{\pm0.001}}$ |
| Traffic | 96 | $\mathbf{0.353_{\pm0.004}}$ | $0.256_{\pm0.001}$ | $0.357_{\pm0.000}$ | $\mathbf{0.251_{\pm0.000}}$ |
| | 192 | $0.372_{\pm0.002}$ | $0.267_{\pm0.000}$ | $\mathbf{0.369_{\pm0.000}}$ | $\mathbf{0.260_{\pm0.000}}$ |
| | 336 | $0.388_{\pm0.002}$ | $0.276_{\pm0.001}$ | $\mathbf{0.383_{\pm0.001}}$ | $\mathbf{0.269_{\pm0.001}}$ |
| | 720 | $\mathbf{0.417_{\pm0.002}}$ | $0.290_{\pm0.001}$ | $0.420_{\pm0.000}$ | $\mathbf{0.287_{\pm0.000}}$ |
| Solar | 96 | $\mathbf{0.175_{\pm0.003}}$ | $0.245_{\pm0.004}$ | $0.182_{\pm0.008}$ | $\mathbf{0.235_{\pm0.006}}$ |
| | 192 | $\mathbf{0.197_{\pm0.001}}$ | $0.262_{\pm0.002}$ | $0.200_{\pm0.003}$ | $0.258_{\pm0.003}$ |
| | 336 | $0.211_{\pm0.002}$ | $0.272_{\pm0.003}$ | $\mathbf{0.209_{\pm0.002}}$ | $\mathbf{0.262_{\pm0.002}}$ |
| | 720 | $0.216_{\pm0.001}$ | $0.275_{\pm0.003}$ | $\mathbf{0.210_{\pm0.000}}$ | $\mathbf{0.264_{\pm0.001}}$ |

Table 20: Results (MSE and MAE) across datasets and horizons. Best mean results are in **bold**, second best are underlined. A dash (–) marks experiments beyond our computational budget.

| Model | H | Electricity | | Weather | | Traffic | | Solar | |
|---|---|---|---|---|---|---|---|---|---|
| | | MSE | MAE | MSE | MAE | MSE | MAE | MSE | MAE |
| MLP + sp. attn. | 96 | $0.140_{\pm0.001}$ | $0.238_{\pm0.001}$ | $0.157_{\pm0.000}$ | $0.202_{\pm0.001}$ | $0.435_{\pm0.006}$ | $0.275_{\pm0.001}$ | $0.201_{\pm0.009}$ | $0.246_{\pm0.003}$ |
| | 192 | $0.167_{\pm0.002}$ | $0.263_{\pm0.002}$ | $0.206_{\pm0.001}$ | $\mathbf{0.249_{\pm0.001}}$ | $0.454_{\pm0.008}$ | $0.286_{\pm0.003}$ | $0.237_{\pm0.003}$ | $0.272_{\pm0.003}$ |
| | 336 | $0.182_{\pm0.002}$ | $0.281_{\pm0.002}$ | $0.264_{\pm0.001}$ | $0.292_{\pm0.001}$ | $0.472_{\pm0.007}$ | $0.292_{\pm0.004}$ | $0.266_{\pm0.006}$ | $0.294_{\pm0.005}$ |
| | 720 | $0.209_{\pm0.008}$ | $0.305_{\pm0.006}$ | $0.349_{\pm0.001}$ | $0.345_{\pm0.001}$ | $0.523_{\pm0.003}$ | $0.316_{\pm0.003}$ | $0.266_{\pm0.004}$ | $0.304_{\pm0.007}$ |
| Pyraf. + sp. attn. | 96 | $0.139_{\pm0.001}$ | $0.236_{\pm0.001}$ | $0.157_{\pm0.002}$ | $0.204_{\pm0.001}$ | $\mathbf{0.389_{\pm0.002}}$ | $0.267_{\pm0.001}$ | $0.188_{\pm0.002}$ | $0.235_{\pm0.003}$ |
| | 192 | $0.157_{\pm0.000}$ | $0.254_{\pm0.000}$ | $0.207_{\pm0.001}$ | $0.251_{\pm0.002}$ | $\mathbf{0.410_{\pm0.001}}$ | $\mathbf{0.276_{\pm0.002}}$ | $0.234_{\pm0.003}$ | $0.270_{\pm0.008}$ |
| | 336 | $0.174_{\pm0.002}$ | $0.273_{\pm0.002}$ | $0.267_{\pm0.001}$ | $0.294_{\pm0.002}$ | $\mathbf{0.420_{\pm0.001}}$ | $\mathbf{0.283_{\pm0.001}}$ | $0.248_{\pm0.001}$ | $0.281_{\pm0.004}$ |
| | 720 | $\mathbf{0.194_{\pm0.001}}$ | $\mathbf{0.292_{\pm0.001}}$ | $0.348_{\pm0.002}$ | $0.347_{\pm0.000}$ | $\mathbf{0.449_{\pm0.002}}$ | $\mathbf{0.300_{\pm0.001}}$ | $0.249_{\pm0.001}$ | $0.280_{\pm0.002}$ |
| iTransformer | 96 | $0.148_{\pm0.000}$ | $0.241_{\pm0.000}$ | $0.171_{\pm0.001}$ | $0.210_{\pm0.001}$ | $0.393_{\pm0.001}$ | $\mathbf{0.266_{\pm0.001}}$ | $0.208_{\pm0.003}$ | $0.240_{\pm0.006}$ |
| | 192 | $0.166_{\pm0.001}$ | $0.259_{\pm0.001}$ | $0.221_{\pm0.001}$ | $0.255_{\pm0.000}$ | $0.424_{\pm0.001}$ | $0.286_{\pm0.001}$ | $0.233_{\pm0.003}$ | $0.257_{\pm0.003}$ |
| | 336 | $0.179_{\pm0.002}$ | $0.274_{\pm0.002}$ | $0.276_{\pm0.000}$ | $0.295_{\pm0.000}$ | $0.430_{\pm0.001}$ | $0.289_{\pm0.001}$ | $0.244_{\pm0.000}$ | $0.273_{\pm0.002}$ |
| | 720 | $0.218_{\pm0.000}$ | $0.309_{\pm0.001}$ | $0.353_{\pm0.000}$ | $0.346_{\pm0.000}$ | $0.455_{\pm0.001}$ | $0.303_{\pm0.000}$ | $0.255_{\pm0.002}$ | $0.280_{\pm0.001}$ |
| Crosformer | 96 | $\mathbf{0.136_{\pm0.000}}$ | $\mathbf{0.232_{\pm0.001}}$ | $\mathbf{0.152_{\pm0.003}}$ | $0.222_{\pm0.004}$ | $0.527_{\pm0.002}$ | $0.270_{\pm0.003}$ | $\mathbf{0.184_{\pm0.008}}$ | $0.227_{\pm0.006}$ |
| | 192 | $0.161_{\pm0.003}$ | $0.257_{\pm0.003}$ | $\mathbf{0.200_{\pm0.007}}$ | $0.271_{\pm0.007}$ | $0.541_{\pm0.002}$ | $\mathbf{0.276_{\pm0.002}}$ | $\mathbf{0.207_{\pm0.015}}$ | $\mathbf{0.246_{\pm0.003}}$ |
| | 336 | $0.187_{\pm0.007}$ | $0.286_{\pm0.007}$ | $0.263_{\pm0.001}$ | $0.325_{\pm0.003}$ | $0.561_{\pm0.005}$ | $0.289_{\pm0.002}$ | $\mathbf{0.222_{\pm0.016}}$ | $\mathbf{0.252_{\pm0.001}}$ |
| | 720 | – | – | $0.359_{\pm0.007}$ | $0.389_{\pm0.004}$ | – | – | $\mathbf{0.214_{\pm0.001}}$ | $\mathbf{0.250_{\pm0.004}}$ |
| ModernTCN | 96 | $0.141_{\pm0.000}$ | $0.237_{\pm0.001}$ | $0.154_{\pm0.001}$ | $\mathbf{0.200_{\pm0.001}}$ | $0.445_{\pm0.001}$ | $0.287_{\pm0.001}$ | $0.190_{\pm0.001}$ | $\mathbf{0.222_{\pm0.002}}$ |
| | 192 | $\mathbf{0.156_{\pm0.001}}$ | $\mathbf{0.249_{\pm0.001}}$ | $0.201_{\pm0.002}$ | $0.252_{\pm0.002}$ | – | – | $0.235_{\pm0.001}$ | $0.250_{\pm0.002}$ |
| | 336 | $\mathbf{0.172_{\pm0.003}}$ | $\mathbf{0.263_{\pm0.001}}$ | $\mathbf{0.254_{\pm0.003}}$ | $\mathbf{0.291_{\pm0.002}}$ | – | – | $0.258_{\pm0.004}$ | $0.266_{\pm0.003}$ |
| | 720 | $0.201_{\pm0.000}$ | $\mathbf{0.292_{\pm0.000}}$ | $\mathbf{0.338_{\pm0.005}}$ | $\mathbf{0.343_{\pm0.005}}$ | – | – | $0.262_{\pm0.003}$ | $0.277_{\pm0.004}$ |

