# OpenReview forum: "What Matters in Deep Learning for Time Series Forecasting?"
_ICLR.cc/2026/Conference — Submitted to ICLR 2026_

### Official Review · Reviewer_uehw · 2025-10-15

**Soundness:** 3
**Presentation:** 3
**Contribution:** 2
**Rating:** 4
**Confidence:** 3

**Summary:**

The authors classifies design choices of time series forecasting models, finding that overlooked details might change the class of forecasting method and have an impact on experiment results. The authors call for future work to use an auxiliary forecasting model card for key design choices. The authors find that, (1) channel hybrid/global impacts model performance; (2) preprocessing would have an impact on time series benchmarkds; (3) no single model outperforms other models, questioning whether temporal model designing are important; (4) show that some spacial model design would give similar results, thus questioning the importance of spacial design.

**Strengths:**

1. The paper calls for rethinking the benchmarks of time series forecasting domain, which I recognize is indeed very necessary and very important.
2. The authors calls for better understanding of architecture's designing space, which might be a method to solve the phenomena that time series forecasting community has been making little progress in the past years.

**Weaknesses:**

1. How are you sure that it's the `model card` rather than the `dataset and benchmarks` that have gone wrong? **Imagine that the CV community are using MNIST rather than CIFAR, ImageNet or other datasets, perhaps researchers could also publish hundreds of papers per year proposing all kinds of CNN/Transformer designs persuing $0.1\%$ improvement on MNIST**. "Oh, my method classifies MNIST better than existing sota". **In that case, you could also do experiments and find "hey, perhaps using vit is similar as convnets, perhaps swinTF is similar to ViT"**. In this case, it is **rethinking, reusing, retargeting datasets and benchmarks** that would help with the problem, rather than making some model cards. Actually, very recently there has been work implying that some datasets for TSF might have gone saturated, for example (https://www.arxiv.org/abs/2510.02729; this paper is online Octobor this year and does not count to my down-rating your paper, but it contributes to my argument that perhaps it's the dataset and benchmarks that have gone wrong.). What's your opinion on this?
2. There have been previous calls for rethinking and utilizing better and more robust time series forecasting and benchmarking. For example, NeurIPS 24 time series in the age of large models workshop Invited Talk by Christoph Bergmeir - Fundamental limitations of foundational forecasting models: The need for multimodality and rigorous evaluation (https://cbergmeir.com/talks/neurips2024/), and also some papers (for example, https://arxiv.org/abs/2502.14045). More recently, some researchers also raise concerns like the time series benchmarks have been saturated. (see weakness 1) **Perhaps the ultimate errors appear in the dataset and benchmarks we are using**, **not in the methods.** Of-course I'm not saying that the methods we propose are fine: saturated datasets and benchmarks might be misleading, resulting in not-that-effective methods. I'm saying that perhaps the dataset issues should be solved first.

**Questions:**

Because a part of this paper seems similar to position paper, I would ask some related questions on behalf of a position paper reviewer, as listed:

Actually I agree that the time series forecasting area has gone wrong in the past several years. Perhaps we should stop overfitting those small simple naive datasets. Do you have some suggestions, advices or opinions on these?

I'm looking forward to further discussions, and I'm potentially willing to increase my score.

---

> ### Author Response · Authors · 2025-11-18
>
> We thank the reviewer for recognizing the importance of our work and appreciate the feedback. We hope to provide all the requested clarifications below.
> >Summary and W1: How are you sure that it's the model card rather than the dataset and benchmarks that have gone wrong?
>
>  We believe there might be a subtle but crucial misunderstanding regarding the main goal of our work. The reviewer argues that our paper suggests that no model outperforms the others and that different architectural designs do not have an impact on performance: this is not the case, nor is it the objective of the paper. Our paper aims to assess whether current **benchmarking and evaluation practices** are truly able to identify the factors contributing to reported performance gains and to disentangle the impact brought by orthogonal design dimensions. We achieve this by showing how important architectural choices – which have often been treated as secondary implementation details – can significantly impact results.  Our paper calls for **adopting more robust and transparent practices in the empirical evaluation**. The purpose of the forecasting model card is precisely to highlight **key** design dimensions. To reiterate, the purpose of our work is not to argue if current models are better than standard baselines, but rather to question whether we, as a community, are relying on effective evaluation practices.
>
> >W2 and Q1: Perhaps the ultimate errors appear in the dataset and benchmarks we are using, not in the methods.
>
> Regarding your questions: are the current datasets saturated? Shouldn’t we focus on finding better datasets? These are important, but **orthogonal** questions. We agree with the reviewer that identifying and proposing new **datasets for benchmarking** is an important aspect, but it is complementary to the discussion presented in the paper. It is the “other side of the coin” that must be addressed to advance the field. We already acknowledge this in the ‘Discussion’ section of the paper (Sec. 5), where we highlight the need for new benchmarks – based on either real or synthetic data – to assess how different design choices impact final performance. Our paper, however, focuses on showing that the approach we have been following to compare different architectures is brittle and unreliable and which can lead to drawing wrong conclusions **irrespective of the datasets being used**. Put differently, even with better and more interesting datasets, the issues we raise in the paper should be taken into account to correctly evaluate different models and ensure reproducible results. Alongside papers like [1] and [2], our work is therefore complementary, addressing aspects that are equally
> crucial to ensure that progress is made. We will clarify this point further in the paper, improving the discussion and making sure that the main focus of the paper is clearer.
>
> We hope that our responses have clarified any points of uncertainty, and we hope that our responses make the scope and contribution of the paper even more transparent. We remain fully available for any further discussion or clarification.
>
> [1] Wang et al., “Accuracy Law for the Future of Deep Time Series Forecasting” arXiv 2025
> [2] Brigato et al., “Position: There are no Champions in Long-Term Time Series Forecasting” arXiv 2025

---

> ### Comment · Reviewer_uehw · 2025-11-18
>
> I'd like to thank the authors for their timely response. However, my concern remains.
>
> I agree with the author that there have been so-called a bit overlooked(they are noticed in previous work, but not mainly discussed in main papers in most previous work) design choices, for example, hybrid/global/local, exogenous vars, etc., and they could have impact to the final result. The authors, therefore, claims that clearly stating and comparing these settings are important and crucial for the development of time series methods.
>
> However, I **don't think the proposed method targeting to resolve this issue is inspiring or novel**. You are basically claiming "Since these hyper-parameter choices are important, we have to clearly compare and do ablation studies on these hyper-parameters." This is intuitive, and the fact that the hyperparameters would have impact to experiment results have been well-known. (e.g. in [2] in your rebuttal reference). I think the "hybrid/global/local", "w./w.o. exogenous vars", etc. shown in your paper simply adds to this fact previously discovered by others. Therefore, I don't think this is of a huge contribution.
>
> Nevertheless, I can get part of your meaning, like "not only architectures are important, hyper-parameter and design choices targeting different datasets are also important". However, related discussions in your paper is not systematic enough or informative enough, nor can it provide great insight to the development of time series forecasting methods. I can easily come up with some ideas or better solutions targeting these difficulties, for example, to call for TSF community to do research like meta-learning (i.e. find methods to obtain the best hyper-parameter and best architecture), to call for TSF community to study the paradigm to get the best methods, etc., but these are not discussed here in your paper. **The authors just go from "results are sensitive to hyperparameters"(a known result of previous work) to "hyperparameter choices are important" and like "we should focus on these hyperparameters with something like model-card", which is intuitive and straightforward**, but you are not discussing more steps forward, for example, how to get these model-cards? Are there possibly better researching paradigms of the TSF community? Can we learn from meta-learning? etc. Therefore, I don't think your work makes great contribution.

---

> > ### Author Response · Authors · 2025-11-23
> >
> > Thank you for your answer, but we respectfully disagree with your assessment of the contributions of our work. Beyond some possible misunderstandings (see below), we agree that the evaluation pitfalls discussed in the paper could in principle be avoided by following certain good practices in time series analysis and, as we note, by more carefully assessing the impact of each design choice. However, this does not reduce the relevance of our findings. Indeed, we believe that showing how current benchmarking practices in the community consistently lead to incorrect conclusions on several aspects is an important contribution. As discussed in the paper, similar analyses have been pivotal in other areas of machine learning, and the corresponding papers have been highly cited and influential within their communities. For example, see [4], published at ICLR, which highlights pitfalls in the evaluation of GNNs. See additional point-by-point answers below.
> >
> > >You are basically claiming "Since these hyper-parameter choices are important, we have to clearly compare and do ablation studies on these hyper-parameters." This is intuitive, and the fact that the hyperparameters would have impact to experiment results have been well-known. (e.g. in [2] in your rebuttal reference).
> >
> > Our work does not concern hyperparameter tuning, but rather the impact of specific design choices and how those reflect on the evaluation practices of recent forecasting architectures. The main contribution is a systematic examination of model configurations and evaluation practices, rather than hyperparameter tuning.
> >
> > >I think the "hybrid/global/local", "w./w.o. exogenous vars", etc. shown in your paper simply adds to this fact previously discovered by others. Therefore, I don't think this is of a huge contribution.
> >
> > We argue that several results in the paper are indeed surprising and significant. For example, 1) seeing a standard MLP or RNN obtain analogous performance to the current state of the art, or 2) improving performance by removing from a state-of-the-art architecture the main components that said architecture was introducing (e.g., spatial attention in iTransformer). Our comprehensive evaluation unequivocally shows many concerns regarding recently reported progress in the field and pinpoints shortcomings of commonly adopted benchmarking practices. As already argued, we firmly believe that this work is a significant contribution for a conference paper.
> >
> > >Nevertheless, I can get part of your meaning, like "not only architectures are important, hyper-parameter and design choices targeting different datasets are also important".
> >
> > Again, we stress that there might be a misunderstanding, since we did not discuss hyperparameter tuning at all.
> >
> > >However, related discussions in your paper is not systematic enough or informative enough, nor can it provide great insight to the development of time series forecasting methods.
> >
> > For our analysis, we consider widely adopted benchmarks for time series forecasting in ML, as well as models recently published in top venues. The datasets and baselines we select are widely used by the community and cover several core aspects of current machine learning for time series forecasting research. **For each dataset and baseline**, we conducted an **extensive hyperparameter search**. We strongly believe this broad and thorough empirical evaluation is rigorous, as acknowledged by other reviewers. The multiple ablation studies and tables support our discussion throughout the paper. We believe these results unequivocally and objectively support our claims. Can you clarify what you mean by not informative enough? We genuinely do not understand how these results and their discussion are not important for the progress of the field.
> >
> > >I can easily come up with some ideas or better solutions targeting these difficulties, for example, to call for TSF community to do research like meta-learning (i.e. find methods to obtain the best hyper-parameter and best architecture), to call for TSF community to study the paradigm to get the best methods, etc., but these are not discussed here in your paper. The authors just go from "results are sensitive to hyperparameters" to "hyperparameter choices are important"
> >
> > As already stated, we think there might be a misunderstanding here regarding the scope of the paper.

---

> > > ### Author Response · Authors · 2025-11-23
> > >
> > > >and like "we should focus on these hyperparameters with something like model-card", which is intuitive and straightforward, but you are not discussing more steps forward, for example, how to get these model-cards?
> > >
> > > The model card is already provided in the Discussion section of the paper (see Sec. 5). We also reference [3] for a general understanding of the use of model cards. Additionally, Appendix D (‘Model cards’) provides a concrete use case. What do you mean by “how to get these model-cards”?
> > >
> > > We hope our response addresses any misunderstandings that may have arisen, and we are happy to discuss more if needed.
> > >
> > >
> > > [3] Mitchell et al. ‘Model cards for model reporting’. FAccT, 2019
> > > [4] Errica et al. ‘A Fair Comparison of Graph Neural Networks for Graph Classification’. ​​ICLR, 2020

---

> ### Comment · Reviewer_uehw · 2025-11-24
>
> I would like to thank the authors for further comments. I apologize for any infromation wrongly converyed between our discussion, but what I mean by "hyperparameter tuning" also includes choices like "hybrid/global/local", "w./w.o. exogenous vars", etc. These are considered some sort of hyperparameter for me too.
>
> I am not claiming that your experiments are not rigorous or not sound; i.e. I have rated the Soundness score to 3: good in my original review.
>
> What I am concerned is mainly about your contribution. I.e. the importance of hyperparameter tuning is discussed and shown in some previous work, and for me I consider things like "hybrid/global/local", etc., as a kind of hyper-parameter choice. For example, for hyperparameter searching as discussed in some Hyperparameter search work, they also consider things like discrete choices or architecture choices as hyperparameter[1,2]. Therefore, your conclusion that these things are important are consistent with the previous findings that hyperparameters are important, or at are only a bit supplementary to existing work.
>
> Based on these, I rated your contribution as 2: fair, overall score as 4: weak reject, and with confidence 3: moderately confident. Personally, I think I understand your paper and I am not miss-understanding central parts of your work or your claims.
>
> **Overall, I would like to thank the authors for these discussions. I recognize that the authors have made rigoros experiments and have written a well-presented and rigoros paper on an interesting topic, but for me I don't think the contribution is above the bar.** I have tried to understand what the authors explained but cannot fully agree with the authors. **Nevertheless, because I am not the author, there might be some parts not completely understood by me. I would lower my confidence from 3 to 2 if you insist that these parts are related to your core contributions and core arguments.**
>
>
>
> Reference:
>
> [1] Hyper-Parameter Optimization: A Review of Algorithms and Applications, arXiv preprint, https://arxiv.org/abs/2003.05689.
>
> [2] Hyperparameter Optimization: Foundations, Algorithms, Best Practices and Open Challenges, Wiley Interdisciplinary Reviews: Data Mining and Knowledge Discovery, https://arxiv.org/abs/2107.05847.

---

> > ### Author Response · Authors · 2025-11-24
> >
> > We believe the discussion is going off track. It is not a matter of what we can or cannot consider a hyperparameter. Our work shows a problem that we believe is very relevant: current benchmarking practices have been ineffective in correctly comparing different architectures and in attributing performance gains to the components being introduced. These problems stem from how certain designs have been incorporated in existing architecture without assessing their actual impact. This has nothing to do with hyperparameter tuning and remains an important finding, regardless of what one might believe is a hyperparameter or not. As already discussed, similar work on the assessment of evaluation procedures has been published at ICLR and fits with the main track criteria for publication.

---

> ### Comment · Reviewer_uehw · 2025-11-26
>
> I would like to thank the authors' keen efforts in rebuttal. However, I personally would not like to change my score.
>
> I have checked some of your referenced paper at ICLR, such as [1]. As stated in [1]'s AC's review, 'position papers can be published at ICLR if it has sufficient novelty and value for the ICLR community': but personally I don't think your paper has made enough contribution to the research community. I.e. It's a well-known fact that the time series community has been making less and less progress on benchmarks, these benchmarks have been (at least partially) saturated, and thus hyperparameters and the so-called design choices (i.e. what the authors argue) would have relatively large impact to the results. This has been a well-known fact and I personally don't think is novel or making huge contribution.
>
> The authors claim that existing papers are not following a good practice. **This could be of great impact if the authors were the first to discover this: unfortunately, they are not. In recent 1-2 years** there have been researchers arguing the practices of time series forecasting community is not that right, and previous work like [2] has shown similar things like hyper-parameters are important, where your work is only supplementary to these pieces of work for me. Therefore I made such an evaluation.
>
> Finally, I have read authors' discussions with other reviewers, and I'd like to say that I am assessing your work from a position paper perspective. Even from this perspective, I don't think your paper meets the standard of 'enough contribution to the research community', or the standard of acceptance. Like I have said, in response to your insisting that I have mis-understood the main part of your work (though I don't think so) and your efforts made in the rebuttal period, I have lower my confidence from 3 to 2; however, I have decided that I will not increase my rating for these concerns listed above.
>
> Reference:
>
> [1] The Alignment Problem from a Deep Learning Perspective, ICLR 2024, https://openreview.net/forum?id=fh8EYKFKns.
>
> [2] Position: There are no Champions in Long-Term Time Series Forecasting, arxiv, https://arxiv.org/abs/2502.14045.

---

### Official Review · Reviewer_HBak · 2025-10-30

**Soundness:** 2
**Presentation:** 2
**Contribution:** 1
**Rating:** 2
**Confidence:** 3

**Summary:**

This paper empirically shows that implementation details (local/global configuration, preprocessing, covariates) have larger impact than architecture choice (Transformer vs MLP) in time series forecasting. Simple baselines match SOTA when properly configured, exposing inconsistent benchmarking practices across recent work.
Contribution: No novel methods—purely empirical analysis exposing benchmarking flaws. Proposes a "model card" template to standardize future comparisons.

**Strengths:**

- Timely and important: Addresses fundamental benchmarking issues affecting the entire time series forecasting community.
- Rigorous empirical work: Comprehensive ablation studies with controlled comparisons across multiple design dimensions.
- Actionable template: The forecasting model card could standardize future research and improve reproducibility.

**Weaknesses:**

- Given the paper's broad claims about deep learning for time series forecasting, the experimental scope (4 datasets, long-range forecasting only, no probabilistic forecasting) seems insufficient to support such general conclusions.
- While experienced practitioners may anticipate some findings (e.g., that preprocessing matters), the systematic quantification of these effects is valuable. However, the paper lacks surprising insights that would significantly advance our understanding.
- The paper is more like a position-track paper, or even a benchmark-track paper, than a main track paper. It lacks theoretical insights, and it only raises issues without providing solutions. (I acknowledge that revealing an important issue is important to the community but the contribution of this paper slightly diverges from what we expect for a main track paper).
- The usefulness of the proposed 'model card' template is uncertain.

**Questions:**

- How confident are you that these findings generalize to short-term forecasting, probabilistic forecasting, and other domains (e.g., irregular time series, multivariate forecasting with true cross-variable dependencies)?
- Beyond diagnosing problems and proposing the model card template, what concrete steps do you think should the community take?

---

> ### Author Response · Authors · 2025-11-18
>
> We thank the reviewer for appreciating the timeliness and importance of our work, as well as for finding our empirical analysis rigorous. We provide point-by-point answers to the raised concerns below.
>
> >W1 and Q1: The experimental scope seems insufficient to support such general conclusions. How confident are you that these findings generalize to short-term forecasting, probabilistic forecasting, and other domains?
>
> For our analysis, we consider widely adopted benchmarks for time series forecasting in ML, as well as models recently published in top venues. The datasets and baselines that we select cover several core aspects of current machine learning for time series forecasting research. Architectures and sequence modeling operators analogous to those discussed here are also used in the contexts mentioned by the reviewer (short-term forecasting, probabilistic forecasting, irregular time series, and more), and the datasets we use are routinely used by the community. Our comprehensive evaluation unequivocally **highlights many concerns regarding recently reported progress in the field** and pinpoints the shortcomings of commonly adopted benchmarking practices. We do this by considering point forecasting as the reference task, but the same architecture and designs are also used in probabilistic, short-term, and other forecasting settings. Our analysis is largely orthogonal to the actual forecasting setting, and we argue that using point forecasting as a reference does not weaken our claims nor make them less concerning. Indeed, our objective is not to show which architecture performs best in a specific task, but rather how certain shortcomings in evaluation can lead to wrong conclusions and unreliable performance assessment. We agree that experiments could be arbitrarily extended to additional settings – including probabilistic and short-term forecasting – but we argue that considering a single reference task is a reasonable choice in the scope of a conference paper. Nonetheless, **we expanded our empirical evaluation** to include results on additional forecasting horizons and window sizes, and extended ablations to a wider set of baselines. You can refer to the change list in the general comment for a complete list of the additional results in the updated version. Please let us know if you have any further feedback on this aspect.
>
> >W2: The paper lacks surprising insights that would significantly advance our understanding.
>
> We agree that some of the pitfalls of current practices are quite evident. However, as a field, we have been using these questionable benchmarking practices consistently. Moreover, while the impact of certain design choices might be clear to time series analysis experts, it might be less obvious to machine learning practitioners. The objective of this paper is to collectively reflect on the impact of these practices on the correctness and reliability of the results obtained with the current benchmarking approach. In this regard, we believe that a conference like ICLR is a great fit for having this kind of discussion. Finally, we argue that several results in the paper are indeed surprising. For example, 1) seeing a standard MLP or RNN obtain analogous performance to the current state of the art, or 2) improving performance by removing from a state-of-the-art architecture the main components that said architecture was introducing (e.g., spatial attention in iTransformer).
>
> >W3: The paper is more like a position-track paper, or even a benchmark-track paper, than a main track paper. It lacks theoretical insights, and it only raises issues without providing solutions.
>
> In addition to our response to W2, we would like to note that ICLR does not have separate calls for benchmarks or position papers, as other conferences do. **ICLR explicitly welcomes contributions of this nature in the main track**, as indicated in the call for papers, and we don’t believe that the lack of theoretical results is a ground for rejection. We appreciate that the reviewer characterizes the work as *timely, important, and rigorous*. We believe that these qualities make it particularly appropriate for discussion at the conference in its current scope.

---

> > ### Author Response · Authors · 2025-11-18
> >
> > >W4: The usefulness of the proposed 'model card' template is uncertain.
> >
> > The usefulness of the proposed model card is twofold. First, it serves as a guideline for practitioners by highlighting important design dimensions—such as locality and globality—whose impact is often underestimated. Second, it improves reproducibility by helping users fully understand the characteristics of the model and the scope of reported results, reducing the time and effort required to extract such information from code or supplementary materials. Indeed, these important design choices are frequently treated as implementation details, as discussed in the paper (see Sections 1 and 4.1). By providing a structured framework for documenting these aspects, the model card promotes transparency, reproducibility, and more informed comparisons across forecasting methods.
> >
> >
> > >Q2: What concrete steps do you think should the community take?
> >
> > As discussed in the paper (see Section 5), we believe that the community should strive to adopt more robust benchmarking practices that account for all the design dimensions and ensure that the empirical evaluation is effective by quantitatively assessing the improvement that the proposed designs bring to the state of the art. The adoption of a clear categorization of the models being compared (e.g., by using model cards) goes toward making this easier. Making evaluation more robust might also require, as a second step, introducing new benchmarks, e.g., by also relying on synthetic datasets. Indeed, when introducing new architectures, authors should aim to identify which components or mechanisms are responsible for observed performance gains and how these relate to the characteristics of the underlying datasets. We believe that acknowledging the impact of the issues discussed in the paper is the first necessary step in this direction. We refer the reviewer to the ‘Discussion’ section of the paper (Sec. 5) for a more detailed discussion on this.
> >
> >
> >
> > We hope that our responses have addressed the reviewer’s concerns. Should there be any remaining questions or points needing clarification, and preventing the reviewer from considering a higher score, we would be happy to continue the discussion.
> >
> > [1] Montero-Manso et al. “Principles and algorithms for forecasting groups of time series: Locality and globality” IJF, 2021
> > [2] Cini, “Taming local effects in graph-based spatiotemporal forecasting”. NeurIPS, 2023

---

> > > ### Comment · Reviewer_HBak · 2025-11-19
> > >
> > > **Summary of Concerns:**
> > > My primary concern is whether this paper's contributions meet the standards for a main track paper at a top-tier venue. While I appreciate the expanded experimental scope, I believe the core contribution is more aligned with a position paper than a main-track paper.
> > >
> > > **Main Issues:**
> > > 1. Limited Novelty of the Central Argument
> > > The paper's main argument—that overlooked design choices significantly impact reproducibility and fair comparison, and should be clearly documented (e.g., via model cards)—is not particularly novel. The importance of transparent reporting of training details and using consistent experimental settings for fair comparisons is already well-established in the research community. Furthermore, prior work such as [1] has already raised similar concerns.
> > > 2. Lack of Actionable Solutions or Deeper Insights
> > > Even granting that the paper identifies a real problem, it does not provide any concrete methodology for it. The proposed solution (model cards/clearer documentation) is essentially a call for better scientific practice rather than a methodological contribution. The paper does not demonstrate that this approach would actually address the identified issues, nor does it provide implementation guidelines or evaluation criteria. If we view this as a methodological contribution, it should be the authors' responsibility to show that it is effective in practice; otherwise, it would be hard to evaluate its usefulness. Moreover, the paper does not make deeper analysis on the proposed problem. The paper does not investigate why these design choices have such significant impact, nor does it provide guidance on which design choices matter most, or how to identify critical hyperparameters in new settings.
> > >
> > > 3. Mismatch with Main Track Evaluation Criteria
> > > Main track papers are typically evaluated on novelty, technical contribution, and clarity. This paper 1) Confirms what practitioners already know through extensive experiments (valuable but not novel), 2) Lacks technical depth—no new methods, algorithms, or theoretical insights, 3) Offers limited guidance for advancing the field beyond 'be more careful and transparent'. It does not even have any actually novel findings that the community would benefit from.
> > >
> > >
> > > **Comparison to Position Papers:**
> > > I want to clarify: I'm not suggesting the paper should have been submitted to a position track, nor that position-style papers cannot be published in main tracks. However, position papers are evaluated differently—on position clarity, significance, and discussion potential rather than technical novelty and methodological contribution. This paper's strengths align more naturally with the position paper criteria, but not with the main track. I would consider giving a higher score if I were reviewing this paper in a position paper track, but here in the main track, I'm sorry I can't raise my score.
> > >
> > > [1] Position: There are no Champions in Long-Term Time Series Forecasting

---

> > > > ### Author Response · Authors · 2025-11-23
> > > >
> > > > Please see our point-by-point answers below.
> > > >
> > > > > Issue 1 & 2: Limited novelty and contribution
> > > >
> > > > As already discussed, our results are orthogonal to, e.g., the results discussed in [1], which mostly concern the quality of the datasets and the impact of hyperparameter tuning. Differently, we focus on issues in the benchmarking practices adopted by the community that result from overlooking core design choices. As we have systematically shown, this can have a strong impact on the observed results and, as we argue, has led to several misleading results.
> > > >
> > > > > The importance of transparent reporting of training details and using consistent experimental settings for fair comparisons is already well-established in the research community.
> > > >
> > > > This is clearly true; we are obviously not saying that the importance of this aspect was unknown in the community. We are instead showing that the practices that we have been routinely following consistently result in misleading outcomes. Model cards are a tool that can make these design choices more evident and, as such, would be useful to practitioners. Again, we do not quite understand what the reviewer would expect us to do more on top of showing the existence of a problem and the effects of this problem on the field. Avoiding the pitfalls described in the paper by making design choices clear as well as assessing their impact on performance would remove the problem. The introduction of model cards is a way to make this easier. Moreover, we argue that making the field aware of these pitfalls and questioning the results we have been relying upon up to this point is already a contribution that would merit discussion at the conference. We believe that the multiple ablation studies and tables support our claims and include insights on the impact of design choices related to the data type (see Sec. 4.3).
> > > >
> > > > > Issue 3: Mismatch with the main track
> > > >
> > > > Again, we do not agree on the fact that the contributions of our paper are not a good match for the main track of a conference like ICLR. Indeed, **analogous papers have been published at ICLR in the past** (see, e.g., [2, 3]) and have had a big impact on the community. As already discussed, we do believe the findings of the paper are novel and relevant. ICLR guide for reviewers states: “Submissions bring value to the ICLR community when they convincingly demonstrate new, relevant, impactful knowledge (incl., empirical, theoretical, for practitioners, etc).” As argued above, we believe our paper fits in such a category.
> > > >
> > > >
> > > > [1] Brigato et al. ‘Position: There are no Champions in Long-Term Time Series Forecasting’. 2025
> > > > [2] Errica et al. ‘A Fair Comparison of Graph Neural Networks for Graph Classification’. ​​ICLR, 2020
> > > > [3] Ngo et al. “The Alignment Problem from a Deep Learning Perspective”. ICLR 2024

---

> > > > > ### Comment · Reviewer_HBak · 2025-11-26
> > > > >
> > > > > 1. What you're arguing in the paper is basically the same point as [1], but you just broadened the scope a bit, which is your novelty but it is far not enough.  Again, spending a massive amount of effort running experiments to prove something that is already well-known to the community does not make your work novel, even if no one else has written that down in a paper. Your argument is not novel, which makes the whole work hardly a contribution to the community.
> > > > >
> > > > > 2. You are claiming that you are providing a method that may solve the issue, but your method is far too naive to be a methodological contribution. Trying to solve the issue basically by calling all researchers to make everything transparent is not a novel approach: it doesn't even count as novelty even as a post on X. A methodological contribution requires 1) novelty, which means others can't come up with this method easily, and 2) empirical effectiveness, i.e., you need experiments or evidence to prove that it works, even on small-scale or synthetic cases. Sadly, what you proposed has neither of these, so this won't count as a contribution from my perspective.
> > > > >
> > > > > 3. Let me clarify this again. I'm not saying that your paper does not match the main track just because it is a position-focused paper. What I'm saying is, **evaluating your paper with a main track standard, it barely has any contribution.** I'm not familiar with the context of [2], but [3] is an early paper that discusses the misalignment problem in a systematic way, which could be valuable, and actually, it was preprinted in Sept. 2022 but accepted by ICLR 2024, which means it should have been rejected in the meantime.
> > > > >
> > > > > In conclusion, **I hold firmly that the contribution of this paper to the community is insufficient for acceptance**. It is a comprehensive evaluation, not merely based on the type of your paper or anything else alone. I would not keep arguing with you as it's a waste of time.

---

### Official Review · Reviewer_eXVU · 2025-11-02

**Soundness:** 2
**Presentation:** 2
**Contribution:** 2
**Rating:** 4
**Confidence:** 4

**Summary:**

The paper examines why deep learning architectures for time series forecasting yield inconsistent and often contradictory results. The authors aim to disentangle the factors influencing model performance and to identify which design elements truly matter in building effective forecasting systems.

The study employs a computational and experimental methodology, combining systematic benchmarking, empirical analysis, and architectural deconstruction. It introduces a framework for analyzing models along four key design dimensions: 1) Model configuration (local, global, hybrid), 2) Preprocessing and exogenous variables, 3) Temporal processing, and 4) Spatial processing.

Through controlled experiments on established benchmarks (Electricity, Weather, Traffic, Solar), the authors compare well-known models such as PatchTST, DLinear, TimeMixer, and Crossformer against a streamlined reference architecture designed to isolate the effects of specific design choices. Key findings include: 1) Many observed performance differences stem from overlooked implementation details—not from architectural innovation. 2) Global or hybrid models, when well-designed, can match or outperform complex state-of-the-art systems. 3) Exogenous variable inclusion and consistent preprocessing have a greater effect on performance than model type. 4) Spatial attention mechanisms contribute little to long-horizon forecasting accuracy.

**Strengths:**

This paper offers a meta-analytical and diagnostic contribution rather than a new predictive model. Its novelty lies in articulating a unified conceptual framework for analyzing deep time series forecasting architectures and demonstrating that benchmarking inconsistencies, not model innovation, explain many reported performance gains. The introduction of a forecasting model card is a valuable proposal for standardizing model documentation, enhancing reproducibility and interpretability across studies.

**Weaknesses:**

- While comprehensive, the study focuses solely on deterministic point forecasting. This leaves out probabilistic and uncertainty-aware approaches, which are central to modern time series applications. The authors acknowledge this but could have elaborated on how their findings generalize to probabilistic settings.

- Although the paper references major forecasting works, it under-engages with recent multimodal and foundation time series models (e.g., TFT, Chronos, pretrained time-series transformers) that might challenge or nuance its conclusions about architecture complexity.

- Most comparisons are run with almost the same look-back window (W) and forecast horizon (H). In fact, they use W=96 for most tables (one table uses W=336, Solar excluded) and H=96 almost always. If we widen the settings (e.g., W=336–720; H=192–336), the relative strengths of architectures can change, so current conclusions may be setting-dependent.

- For spatial models, the authors explicitly shrink W to 96 “to keep costs manageable,” and then conclude spatial attention adds limited value. But some cross-series patterns can only emerge with longer windows/horizons; the constraint itself may handicap spatial operators.

**Questions:**

- If we sweep W ∈ {96, 336, 720} and H ∈ {96, 192, 336}, do the two core claims still hold: (i) simple models match SOTA, and (ii) spatial attention helps little? Where do rankings flip as context grows? (This matters because current runs mostly fix W=96 and H=96.)

- Under probabilistic evaluation, do simpler models still lead? If not, the paper’s guidance should be framed as point-forecast-specific.


- Model configuration (Local, Global, Hybrid) represents a core design choice that directly affects model behavior and capacity. Why do the authors argue that configuration effects should be factored out rather than treated as part of each model’s intended design?
Would controlling for configuration risk removing meaningful aspects of a model’s inductive bias and thereby alter its intended behavior?

- Table 1 compares models under Hybrid and Global setups, yet the paper does not clearly explain how each version was implemented.
What exactly was modified in each model to create the “Global” or “Hybrid” configuration (e.g., were per-series normalization parameters or embeddings added/removed)? Providing this procedural detail would make the comparison reproducible and easier to interpret.

- Is the proposed way of constructing a Hybrid model (shared parameters plus per-series components) a general framework that can be applied to any architecture, or is it specific to certain models like TimeMixer and Crossformer?
Clarifying this would help readers understand whether Hybrid configuration is a standardized recipe or an ad-hoc adjustment.

- Could similar experiments be performed under a Local configuration (one model per time series) for at least a subset of the deep models?
If not, could the authors discuss the practical or computational reasons preventing this?
Such results would help establish a complete Local–Global–Hybrid comparison.

- Linear baselines (e.g., Linear, DLinear) can in principle be trained under different configurations.
Is it feasible to evaluate these models under Hybrid or Global settings as well?
Including these variants could strengthen the paper’s conclusions by showing whether configuration effects are consistent across both deep and linear models.

---

> ### Author Response · Authors · 2025-11-18
>
> We thank the reviewer for their thoughtful and constructive feedback. We provide point-by-point answers to the raised concerns below.
>
> >W1 and Q2: The study focuses solely on deterministic point forecasting. The authors acknowledge this but could have elaborated on how their findings generalize to probabilistic settings. Under probabilistic evaluation, do simpler models still lead?
>
> For our analysis, we consider widely adopted benchmarks for time series forecasting in ML, as well as models recently published in top venues. The datasets and baselines that we select cover several core aspects of current machine learning for time series forecasting research. Architectures and sequence modeling operators analogous to those discussed here are also used in probabilistic forecasting and other contexts (short-term forecasting, irregular time series, and more), and the datasets we use are routinely used by the community. Our comprehensive evaluation unequivocally **highlights many concerns regarding recently reported progress in the field** and pinpoints the shortcomings of commonly adopted benchmarking practices. We do this by considering point forecasting as the reference task, but the same architecture and designs are also used in probabilistic, short-term, and other forecasting settings. Our analysis is largely orthogonal to the actual forecasting setting, and we argue that using point forecasting as a reference does not weaken our claims nor make them less concerning. Indeed, our objective is not to show which architecture performs best in a specific task, but rather how certain shortcomings in evaluation can lead to wrong conclusions and unreliable performance assessment. We agree that experiments could be arbitrarily extended to additional settings – including probabilistic forecasting – but we argue that considering a single reference task is a reasonable choice in the scope of a conference paper. Nonetheless, we **expanded our empirical evaluation** to include results on additional forecasting horizons and window sizes, and extended ablations to a wider set of baselines. You can refer to the change list in the general comment for a complete list of the additional results in the updated version. Please let us know if you have any further feedback on this aspect.
>
> >W2: Although the paper references major forecasting works, it under-engages with recent multimodal and foundation time series models
>
> As the reviewer acknowledges, the main aim of our paper is to assess whether current **benchmarking and evaluation practices** are truly able to identify the factors contributing to reported performance gains and to disentangle the impact brought by orthogonal design dimensions. We would like to emphasize that this is the main focus of the work, rather than comparing models with different levels of complexity. While our study has implications for the design of foundation models, extending the empirical analysis to such approaches is out of scope and would require additional considerations beyond the focus of this paper. Indeed, these models are designed for broad, cross-domain applicability and are typically provided as ready-to-use, while we discuss issues in model selection. As such, it is not clear how foundation models could be assessed in the scope of our study.
>
>
> >W3 and Q1: If we widen the settings, the relative strengths of architectures can change, so current conclusions may be setting-dependent. If we sweep W ∈ {96, 336, 720} and H ∈ {96, 192, 336}, do the two core claims still hold: (i) simple models match SOTA, and (ii) spatial attention helps little? Where do rankings flip as context grows?
>
> For the scope of our analysis, our comprehensive evaluation **already addresses many concerns regarding recently reported progress in the field and pinpoints the shortcomings of commonly adopted benchmarking practices**. Indeed, the scope of the paper is not to provide an exhaustive benchmarking of models, but rather to show how important design choices, often treated as mere implementation details, can drastically alter results and lead to incorrect attribution of performance gains. However, we have added **new tables in Appendix E (Tab. 14–18)** evaluating longer input windows and forecasting horizons to strengthen the conclusions of Sections 4.3 and 4.4. Moreover, to broaden our analysis, we have **extended Tables 1 and 2 (Sec. 4.1 and 4.2)** by including additional models. These updates have already been incorporated into the revised version of the paper. Moreover, **we are currently running additional experiments**, and we will report the remaining results here before the end of the rebuttal period, posting a general comment as soon as they are ready.

---

> > ### Author Response · Authors · 2025-11-18
> >
> > >W4: For spatial models, the authors explicitly shrink W to 96…. But some cross-series patterns can only emerge with longer windows/horizons.
> >
> > For the datasets considered in this work, it is reasonable to assume that cross-series dependencies are useful only for short-range prediction. For example, a look-back window and horizon length of 96 steps in the hourly datasets (Electricity, Traffic) correspond to processing the previous 4 days' readings to predict the next 4 days. In these domains, the influence of neighboring observations (e.g., the car speed measured by other sensors in the same road network) on a target time series clearly decreases with time. Said differently: knowing the traffic conditions at a neighboring intersection might be useful in forecasting what will happen in the next, say, 10 minutes, but would bear little relevance in predicting what will happen in two days from now.
> >
> > >Q3: Why do the authors argue that configuration effects should be factored out rather than treated as part of each model’s intended design?
> >
> > There might be a misunderstanding here. We agree that using a Local/Global/Hybrid method is part of a model design. The point of our discussion is that this design choice — along with all the others discussed in the paper — should be clearly and transparently discussed, and its impact on the performance of the proposed forecasting architecture should be carefully assessed. Failing in doing so may result in attributing differences in performance to other aspects (e.g., the sequence modeling operators being used), which ultimately hinders the understanding of the gain in performance introduced by each component. This is what we mean when we say that this aspect should be “factored out” in the analysis. Moreover, note that global and hybrid models have very different properties (e.g., a global model is inductive), which should be taken into account with respect to the characteristics of the dataset. Indeed, as shown in Table 3 (Sec. 4.3), the local linear model — the only approach explicitly modeling each time series as heterogeneous — is among the best-performing models on the Weather dataset.
> >
> > >Q4 and Q5: What exactly was modified in each model to create the “Global” or “Hybrid” configuration? Is the proposed way of constructing a Hybrid model a general framework that can be applied to any architecture?
> >
> > Our policy for obtaining the global and hybrid variants of models is detailed in the paragraph “Empirical setup for D1: model configuration” (Appendix C); **we report a summary in what follows**. To obtain the global version of a model that already includes local parameters, we remove those local components. For example, the global TimeMixer model is obtained by removing the learnable parameters from the normalization module, while the global versions of Crossformer and the reference architecture are obtained by removing their local embeddings. Conversely, to create a hybrid version of an otherwise global model, we add per-series local embeddings, which are used as an additional input. This approach is straightforward and can be applied to different architectures, as also shown by recent works [1,2].
> >
> > >Q6: Could similar experiments be performed under a Local configuration (one model per time series) for at least a subset of the deep models?
> >
> > Yes, it is theoretically possible to train a local model for each time series, but in practice, this is unfeasible when using a complex model, as it would require training N separate models for N time series.  Prior work ([3,4,5]) has shown that global or hybrid approaches consistently outperform the purely local models, particularly when the time series are homogeneous.

---

> > > ### Author Response · Authors · 2025-11-18
> > >
> > > >Q7: Linear baselines (e.g., Linear, DLinear) can in principle be trained under different configurations. Is it feasible to evaluate these models under Hybrid or Global settings as well?
> > >
> > > We appreciate the reviewer’s suggestion to include hybrid variants of these models. However, Linear and DLinear are linear autoregressive models, and weights directly map inputs to predictions; it is not clear how one would introduce hybrid versions thereof without introducing a substantially different model. However, we agree that it is worth exploring this possibility. Accordingly, **we added an experiment in Appendix E** to assess the performance of two simple hybrid variants of such architectures (see **Table 18**). The results are consistent with the discussion presented in Section 4.1 ('Design Dimension 1: model configuration’).
> > >
> > >
> > > Please let us know if you believe that the proposed edits and newly added experiments are not enough to address your doubts. We are also running further experiments and will provide a general update once the revision and new results are ready.
> > >
> > > [1] Shao et al. "Spatial-temporal identity: A simple yet effective baseline for multivariate time series forecasting." CIKM 2022.
> > > [2] Cini et al. "Taming local effects in graph-based spatiotemporal forecasting." NeurIPS, 2023.
> > > [3] Smyl, "A hybrid method of exponential smoothing and recurrent neural networks for time series forecasting." IJF, 2020.
> > > [4] Salinas et al. "DeepAR: Probabilistic forecasting with autoregressive recurrent networks." IJF, 2020.
> > > [5] Hewamalage et al. "Recurrent neural networks for time series forecasting: Current status and future directions." IJF, 2021.

---

> > > > ### Author Response · Authors · 2025-11-25
> > > > **Additional results**
> > > >
> > > > Dear reviewer, we have updated the paper to include the remaining additional results you suggested (see general comment). We hope that our rebuttal and the additional results addressed all concerns. Please let us know if there is anything else preventing you from recommending a higher score.

---

### Author Response · Authors · 2025-11-18
**New experiments and changelog:**

To address the reviewers’ comments and provide a more comprehensive evaluation, we have added to the paper the following additional results:

## Main Paper
- **Table 1** (**Section 4.1** – 'Design Dimension 1: model configuration’): added **iTransformer** to compare models with and without local parameters.
- **Table 2** (**Section 4.2** – 'Design Dimension 2: preprocessing and exogenous variables’): Added **iTransformer** and **Crossformer** to compare models with and without covariates.

## Appendix (new Section E: Additional results)

We added an extensive set of new experiments to broaden the empirical analysis:
- **Multi-horizon results**: we added **Table 14** to extend the results in Table 3 (Section 4.3 – 'Design Dimension 3: temporal processing’) to longer horizons (96, 192, 336, 720).
- **Extended ablation on iTransformer**: we added **Tables 15, 16, and 17** to expand the original ablation in Table 4.b (Section 4.4 – 'Design Dimension 4: spatial processing’) to additional forecasting **horizons** (96, 192, 336, 720) and **window** sizes (96, 336).
- **Linear autoregressive models**: we added **Table 18** to compare **local, global, and hybrid** variants of Linear and DLinear.

The additional results further support the main claims of the paper and **reinforce our analysis** of Sections 4.3 and 4.4. Results on the linear baselines are also **consistent** with the findings discussed in Section 4.1.


**We are currently running additional experiments** to further broaden the scope of the study. The manuscript will be updated with these results before the end of the rebuttal period, and the reviewers will be notified promptly as soon as they are available.

---

> ### Author Response · Authors · 2025-11-25
>
> # New experiments and changelog - part 2:
> In addition to the experiments reported in the first changelog, we have now included several additional results in the revised version of the paper. The new experiments provide further empirical evidence to support the analysis in Section 4.4.
>
> - **Extended results for spatial processing**:
> We added **Table 20**, which extends the results from **Table 4.a** (Section 4.4) to longer forecasting horizons: 96, 192, 336, and 720.
>
> - **Expanded iTransformer ablation**:
> We added **Table 19**, which expands the original ablation study in **Table 4.b** (Section 4.4) to include a longer window size of 720 and forecasting horizons of 96, 192, 336, and 720. Moreover, we updated **Table 17** with additional results across increasing window sizes under a fixed forecasting horizon.

---

### Author Response · Authors · 2025-12-03
**Final comments**

Dear Area Chair, Dear Reviewers,


We wanted to conclude the discussion with a summary of our position and our answers to the reviewers’ concerns.


Our paper identifies several critical issues in how we, as a community, evaluate deep learning architectures for time series forecasting. These benchmarking practices have led the community to misleading and seemingly contradictory conclusions. We show that the current approach to benchmarking forecasting architectures does not account for how key design choices affect performance, resulting in performance differences instead being attributed to the wrong components (e.g., the sequence modeling operator). This analysis is novel, and though other works debating the current state of the field exist, our work pinpoints specific sources of the inconsistencies often seen in benchmarking results.


Reviewers believe that this is not enough for a main-track conference paper; we disagree and believe that the paper contains insights that would be interesting for the community and useful to the advancement of our field. Moreover, reviewers suggest that our analysis does not provide additional insights over papers showing that most architectures would obtain similar performance with appropriate hyperparameter tuning; again, we disagree, as the problems we discuss are orthogonal to how hyperparameters are tuned.


Finally, we were disappointed with the tone that the discussion took with some of the reviewers; we are obviously open to different views and constructive criticism, but believe that comments comparing a paper to “a post on X” or questioning the validity of a reference since it was accepted two years after preprint have no place in scientific discourse.


Thank you for reviewing our paper.

---

### Meta-Review · Area_Chair_W7G9 · 2026-01-11

**Summary:**

There was a long back-and-forth about whether this should be read as a position/benchmarking piece versus a main-track contribution. Regrettably, some of the discussion could have been carried out in a more respectful manner. But stepping away from that: evaluated as an empirical study, the paper’s scope and experimental design are too limited to support the strength of the conclusions it gestures at. The experiments are careful within a narrow slice of settings, but the paper’s claims generalize beyond what is actually covered (task scope, dataset breadth, regime coverage), so I’m not convinced the study, as executed, can settle the questions it raises.

**Reviewer Concerns:**

The rebuttal did address some concrete points—expanded some settings (e.g., longer windows/horizons), added/clarified comparisons, and explained implementation details around “global/hybrid” variants. What remains is the bigger issue: the empirical coverage is still too limited to justify broad takeaways, key axes are out of scope (and the paper doesn’t sufficiently narrow its claims to match), and the proposed “model card” remedy remains more of a documentation suggestion than a demonstrated fix for the failure modes.

**Reviewer Scores:**

All likely staying put or lower their scores given the shared sentiment of interesting observations, but insufficient evidences to make them stick.

---

### Decision · Program_Chairs · 2026-01-26

Reject